# Activation of an atypical plant NLR with an N-terminal deletion initiates cell death at the vacuole

Sruthi Sunil [ID][1], Simon Beeh [ID][1], Eva Stöbbe[1], Kathrin Fischer[1], Franziska Wilhelm[1], Aron Meral[1], Celia Paris [ID][1], Luisa Teasdale[2], Zhihao Jiang[1], Lisha Zhang [ID][1], Moritz Urban[1], Emmanuel Aguilar Parras[3,5], Thorsten Nürnberger[1], Detlef Weigel[2,4], Rosa Lozano-Duran[1,3] & Farid El Kasmi [ID][1✉]

## Abstract

**Plants evolve nucleotide-binding leucine-rich repeat receptors (NLRs) to induce immunity. Activated coiled-coil (CC) domain containing NLRs (CNLs) oligomerize and form apparent cation channels promoting calcium influx and cell death, with the alpha-1 helix of the individual CC domains penetrating the plasma membranes. Some CNLs are characterized by putative N-myristoylation and S-acylation sites in their CC domain, potentially mediating permanent membrane association. Whether activated Potentially Membrane Localized NLRs (PMLs) mediate cell death and calcium influx in a similar way is unknown. We uncovered the cell-death function at the vacuole of an atypical but conserved Arabidopsis PML, PML5, which has a significant deletion in its $CC_{G10/GA}$ domain. Active PML5 oligomers localize in Golgi membranes and the tonoplast, alter vacuolar morphology, and induce cell death, with the short N-terminus being sufficient. Mutant analysis supports a potential role of PMLs in plant immunity. PML5-like deletions are found in several Brassicales paralogs, pointing to the evolutionary importance of this innovation. PML5, with its minimal CC domain, represents the first identified CNL utilizing vacuolar-stored calcium for cell death induction.**

**Keywords** NLR; Resistosome; Tonoplast; N-myristoylation and S-acylation; Cell Death and Calcium Influx
**Subject Categories** Immunology; Membranes & Trafficking; Signal Transduction

## Introduction

Plant immunity is mediated by two types of immune receptors: pattern-recognition receptors (PRRs) and nucleotide-binding leucine-rich repeat receptors (NLRs) (Ngou et al, 2021). PRRs detect conserved molecular patterns (i.e., pathogen-associated patterns—PAMPs) at the cell surface and transmit this information to the inside of the cell via phosphorylation cascades. The downstream responses include production of reactive oxygen species, transient $Ca^{2+}$ fluxes, and transcriptional reprogramming (Bigeard et al, 2015; Jeworutzki et al, 2010). NLRs sense the activity of pathogen-derived effector proteins (also known as virulence factors) translocated into the host cell. Effector-triggered and NLR-mediated immunity is mostly a potentiation of PRR-mediated immune outputs (Ngou et al, 2021; Yuan et al, 2021). However, NLR activation often also leads to a specialized type of programmed cell death, the hypersensitive response (HR), of the infected cell (Balint-Kurti, 2019; El Kasmi, 2021). NLRs can be classified into four major subclasses based on their N-terminal domains: Toll-like/interleukin 1 receptor (TIR) domain containing TNLs, coiled-coil (CC) domain containing CNLs, RPW8-like CC ($CC_R$) domain containing RNLs, and the ancient and autonomous $CC_{G10/GA}$-type NLRs, or ANLs (Kourelis, 2022; Lee et al, 2021).

Many TNLs, CNLs, and ANLs have been shown to sense effectors directly or indirectly, whereas the two RNL subfamilies, the ACTIVATED DISEASE RESISTANCE 1 proteins (ADR1s) and N REQUIREMENT GENE 1 proteins (NRG1s), also referred to as helper NLRs, are required downstream of all TNLs and some sensor CNLs and ANLs (Castel et al, 2019; Jubic et al, 2019; Saile et al, 2020; Wu et al, 2019). Helper NLR-independent NLRs can autonomously induce immune responses (Maruta et al, 2022). An example is the *Arabidopsis thaliana* (hereafter Arabidopsis) CNL HopZ-ACTIVATED RESISTANCE 1 (ZAR1), which confers resistance to a variety of bacterial pathogens (Baudin et al, 2017). Effector recognition induces conformational changes that allow AtZAR1 monomers to form a pentameric resistosome complex (Wang et al, 2019b; Wang et al, 2019a). The AtZAR1 resistosome functions at the plasma membrane (PM) where the alpha-1 helices at the very N-terminus of each of the five AtZAR1 CC domains penetrate the PM to form a cation-permeable channel important for $Ca^{2+}$ influx. Calcium influx is in turn a prerequisite for AtZAR1-induced disease resistance and cell death (Bi et al, 2021; Wang et al, 2019b). A similar cell-death inducing mechanism has been postulated for the Arabidopsis RNLs AtADR1, AtNRG1.1, and AtNRG1.2, which also form higher-order oligomeric $Ca^{2+}$

[1]Centre for Plant Molecular Biology, University of Tübingen, 72076 Tübingen, Germany. [2]Max Planck Institute for Biology Tübingen, 72076 Tübingen, Germany. [3]Shanghai Center for Plant Stress Biology, CAS Center for Excellence in Molecular Plant Sciences, Chinese Academy of Sciences, Shanghai 200032, China. [4]Institute for Bioinformatics and Medical Informatics, University of Tübingen, 72076 Tübingen, Germany. [5]Present address: Instituto de Hortofruticultura Subtropical y Mediterránea 'La Mayora', Universidad de Málaga-Consejo Superior de Investigaciones Científicas, Depto. Biología Celular, Genética y Fisiología, 29010 Málaga, Spain. ✉E-mail: farid.el-kasmi@zmbp.uni-tuebingen.de

permeable channels at the plasma membrane in their (auto-) activated state (Feehan et al, 2023; Jacob et al, 2021; Wang et al, 2023). The importance of the CC, $CC_R$, and $CC_{G10/GA}$ domains for cell death activity is further supported by the observation that overexpression of these domains alone is often sufficient to induce cell death or autoimmune symptoms (Bolus et al, 2020; Casey et al, 2016; Collier et al, 2011; Jacob et al, 2018; Kim et al, 2018; Lee et al, 2022; Wang and Balint-Kurti, 2015; Wroblewski et al, 2018). Indeed, the first 29 amino acids of the *Nicotiana benthamiana* CNL NLR-REQUIRED FOR CELL DEATH 4 (NRC4), constituting the potentially membrane-penetrating alpha-1 helix, are sufficient to induce weak cell death when fused to YFP and transiently overexpressed (Adachi et al, 2019). Furthermore, exchanging the very N-terminal 17 residues of NRC4, corresponding to approximately half of the alpha-1 helix, with the corresponding region of several other unrelated CNLs revealed a conserved function of this region for CNL cell death activity. This N-terminal region also contains conserved (mostly hydrophobic) residues that are essential for cell death and resistance function of AtZAR1 and NbNRC4 (Adachi et al, 2019).

Oligomerization and resistosome formation have been demonstrated for several cell-death inducing CNLs and RNLs (Ahn et al, 2023; Bolus et al, 2020; Feehan et al, 2023; Forderer et al, 2022; Jacob et al, 2021; Li et al, 2020; Wroblewski et al, 2018), thus leading to the hypothesis that the $CC/CC_R/CC_{G10/GA}$ domains of most CNLs, including RNLs and ANLs, can form membrane-penetrating channels that promote cell death and immunity.

The ANL subclass of CNLs is characterized by the absence or severe degeneration of conserved motifs in the CC domain that have been implicated in cell death activity of other CNLs (Lee et al, 2021). All ANLs that have been studied in detail appear to be (plasma-)membrane localized and their $CC_{G10/GA}$ domains alone can induce a cell death response when overexpressed (Huang et al, 2024; Kourelis et al, 2021; Lee et al, 2021; Wroblewski et al, 2018). The Arabidopsis Col-0 reference genome encodes 23 ANLs (Table EV1), including the well-characterized plasma membrane-localized proteins RESISTANCE TO PSEUDOMONAS SYRINGAE 2 (RPS2) and 5 (RPS5), SUPPRESSOR OF TOPP4-1 (SUT1), SUPPRESSOR OF MKK1 MKK2 2 (SUMM2), and the recently described NLRs, L3 and L5 (Huang et al, 2021; Kourelis et al, 2021; Lee et al, 2021; Seo et al, 2016; Yan et al, 2019; Zhang et al, 2012).

Plasma membrane localization is important for CNL-/ANL- and RNL-dependent immunity and cell death (Axtell and Staskawicz, 2003; Chen et al, 2017; El Kasmi et al, 2017; Huang et al, 2024; Huang et al, 2021; Qi et al, 2012; Saile et al, 2021; Takemoto et al, 2012). 17 of the 23 Arabidopsis ANLs feature residues subject to potential co- and/or post-translational modifications (PTMs) important for their (constitutive) (plasma-)membrane localization (Table EV1). For example, AtRPS5 is potentially N-myristoylated at glycine residues 2 and 3 and S-acylated (S-palmitoylated) at cysteine 4 (Qi et al, 2012). N-myristoylation is thought to be an irreversible modification in which myristic acid is added to the N-terminus of proteins starting with methionine (M)–glycine (G)-X-X-X-serine (S)/threonine (T) residues, whereas S-acylation is a reversible modification usually involving the addition of a palmitate group to any cysteine (C) residue throughout the protein (Wang et al, 2021). S-acylation is thought to regulate membrane localization, protein-protein interactions, and cellular signaling processes (Hurst et al, 2023; Peitzsch and McLaughlin, 1993; Resh,

1999; Wang et al, 2021). Plasma membrane localization and cell death activity are lost in several ANLs, including AtRPS5, AtR5L1, AtL3, and AtL5 when residues G2, G3, and C4 are mutated. This is consistent with a central role of PTMs at these residues (Gao et al, 2022; Huang et al, 2024; Huang et al, 2021; Qi et al, 2012). The extensive conservation of these residues and their importance for cell death activity of ANLs raises an important question regarding the resistosome/cation channel model. How can these NLRs, or more specifically their $CC_{G10/GA}$ domains, trigger cell death and immunity if their very N-terminus, preceding the alpha-1 helix, is permanently associated with the inner leaflet of the plasma membrane?

We have identified and characterized an ANL with a deletion of 113 amino acids at the N-terminus and potentially N-myristoylated glycine G2 and S-acylated cysteines C3 and C4 in the $CC_{G10/GA}$ domain, which we named PML5 (Potentially Membrane Localized 5). PML5 provides a platform to study whether and how such variant ANLs function in cell death initiation and immunity. We demonstrate that PML5 is a canonical, self-associating ANL that forms activation-dependent higher-order oligomers. Overexpression of PML5 causes strong cell death responses when expressed either in *N. benthamiana* or Arabidopsis and leads to an increase in cytosolic calcium concentration possibly by releasing calcium from the vacuole, since localization of (active) PML5 to Golgi membranes and the tonoplast, but not to the cytosol, is required for cell death, which can be induced when only the first 60 amino acids of PML5 ($CC^{1-60}$) are overexpressed. PML5-like N-terminal deletions are also found in ANLs of other Arabidopsis accessions and in several Brassicales. Taken together, our results suggest that potentially constitutively membrane-associated ANLs have a canonical cell death function that requires their predicted alpha-1 helix, and that the PML5-like ANLs may fulfill an important evolutionary conserved function. Furthermore, our results suggest that NLR-triggered cell death and calcium influx may also be mediated at the vacuole.

## Results

### PML5 is an ANL with a conserved 113 amino acid deletion in its $CC_{G10/GA}$ domain

N-terminal CC domains of CNLs, which form plasma membrane-penetrating cation channels, are essential for immune signaling and cell death initiation (Adachi et al, 2019; Bi et al, 2021; Bolus et al, 2020; Forderer et al, 2022; Qi et al, 2012; Wang et al, 2019b; Wang et al, 2019a). In Arabidopsis Col-0, we identified 21 CNLs with putative N-myristoylation and/or S-acylation sites at their N-terminus, which we specifically named Potentially Membrane Localized (PML) NLRs or short PMLs (Table EV1 and Fig. 1A). 17 of these PMLs belong to the monophyletic group of autonomous CNLs with a distinct $CC_{G10/GA}$ domain, termed ANLs (Appendix Fig. S1 and Table EV1) (Kourelis et al, 2021; Lee et al, 2021). Detailed analysis of their coding sequences revealed that one of them, *PML5/At1g61300*, encodes for a 762 amino acid protein with a deletion of 113 amino acids in its $CC_{G10/GA}$ domain (Appendix Fig. S2). PML5 has all canonical motifs in its nucleotide-binding (NB-ARC) domain (Fig. 1B) and forms a small subclade (cluster #3) with PML11, 12, and 13, which are all full-length PMLs

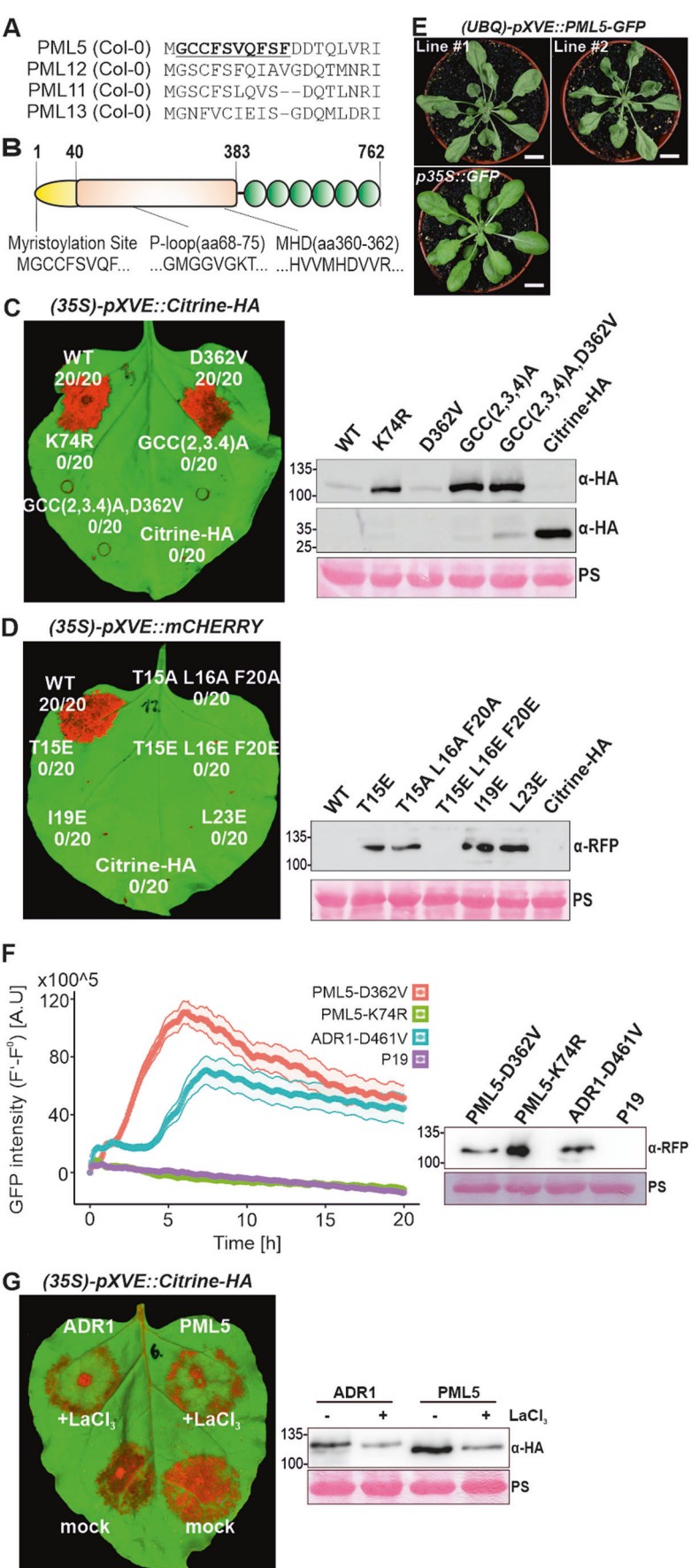

**A**

PML5 (Col-0)  M**GCCFSVQFSF**DDTQLVRI
PML12 (Col-0)  MGSCFSFQIAVGDQTMNRI
PML11 (Col-0)  MGSCFSLQVS--DQTLNRI
PML13 (Col-0)  MGNFVCIEIS-GDQMLDRI

**B**

Myristoylation Site  P-loop(aa68-75)  MHD(aa360-362)
MGCCFSVQF...  ...GMGGVGKT...  ...HVVMHDVVR...

**E**  *(UBQ)-pXVE::PML5-GFP*

Line #1    Line #2

*p35S::GFP*

**C**  *(35S)-pXVE::Citrine-HA*

WT 20/20    D362V 20/20
K74R 0/20    GCC(2,3.4)A 0/20
GCC(2,3.4)A,D362V 0/20    Citrine-HA 0/20

α-HA
α-HA
PS

**D**  *(35S)-pXVE::mCHERRY*

WT 20/20    T15A L16A F20A 0/20
T15E 0/20    T15E L16E F20E 0/20
I19E 0/20    L23E 0/20
Citrine-HA 0/20

α-RFP
PS

**F**

GFP intensity (F¹-F⁰) [A.U]

PML5-D362V
PML5-K74R
ADR1-D461V
P19

Time [h]

α-RFP
PS

**G**  *(35S)-pXVE::Citrine-HA*

ADR1    PML5
+LaCl₃    +LaCl₃
mock    mock

LaCl₃
α-HA
PS

◄ **Figure 1. PML5 functions as a canonical cell death and calcium influx inducing CNL.**

(A) Sequence alignment of the first 23 N-terminal amino acids of PML5 and related NLRs, highlighting the potential N-myristoylation sites in PML5. (B) Scheme of PML5 domain composition. Indicated are the potential S-acylation and N-myristoylation site and the P-loop as well as the MHD motifs. (C, D) Cell death induced by transiently expressed wild-type and mutant PML5 variants in *N. benthamiana* leaves (left) and corresponding protein blots (right). Leaves are shown in false color, red indicates cell death, and green healthy/alive tissue. Photos was taken 3 days after induction with 20 μM estradiol. Proteins were extracted 6 h after induction with 20 μM estradiol and detected with an anti-HA (α-HA) antibody (C) and anti-RFP (a-RFP) antibody (D). Citrine-HA was used as negative control. Bottom blot in (C) shows Citrine-HA (about 30 kDa). Ponceau S (PS) staining is shown as loading control. WT = wild-type PML5; D362V = MHD mutant; K74R = P-loop mutant; GCC(2,3,4)A = N-myristoylation and S-acylation (PTM) mutant; GCC/2,3,4)A,D362V = PTM/MHD quadruple mutant. (E) Cell death in two independent transgenic Arabidopsis lines conditionally overexpressing *PML5-GFP*. A *35::GFP* plant was used as negative control. Four-week-old plants 30 h after induction with 20 μM estradiol are shown. Scale bars, 1 cm. (F) PML5 induces calcium ion influx in GCamp3 transgenic *N. benthamiana*. *PML5-D362V-mCherry*, *PML5-K74R-mCherry*, *ADR1-D461V-mCherry* (positive control), and *P19* (negative control) expressing constructs were agroinfiltrated. Calcium influx was measured over 20 h. Plotted values are averages of 24 leaf discs from 6 leaves of individual plants, with Standard Error represented. The corresponding protein blot (bottom) with anti-RFP (a-RFP) antibody is shown on the right. Total proteins were extracted 4 h post induction with 80 μM estradiol. P19 was used as negative control. Ponceau S (PS) staining is shown as loading control. (G) PML5 cell death is blocked by LaCl₃. 2 mM LaCl₃ were co-infiltrated with *PML5* and *ADR1* Citrine-HA-tagged constructs. Expression was induced 20 h after infiltration and samples for protein blots were collected 5–6 h later. Protein expression was detected with an anti-HA (α-HA) antibody. Ponceau S (PS) staining is shown as loading control. Cell death was imaged 24 h after inducing the expression with 80 μM estradiol. Data information: In (F), data are represented as mean ± SEM. Source data are available online for this figure.

(Appendix Figs. S1 and S2). AlphaFold 2 modeling of PML5 predicted a well-structured NB-ARC and C-terminal LRR domain, while the CC domain was predicted with low confidence as a short unstructured region followed by a single alpha-helix (amino acid 13–29) (Fig. EV1A).

We examined the genomes of other Arabidopsis accessions (Jiao and Schneeberger, 2020; Liu et al, 2021; Van de Weyer et al, 2019) to determine whether other accessions also contain genes with a *PML5*-like deletion. In the genomes of 14 accessions, at6909/Col-0; at9837/Con-0; at9721/Schip-1; at7413/Wil-2; at9762/Etna-2; at1925/Che-2; at7416/Yo-0; at7058/Bur-0; at9814/Fell1-10; at9776/Fell3-7; at8285/Dralll-1; at6898/An-1; at7207/Kyoto; at9583/IP-Sne-0, a gene with high similarity to Col-0 *PML5*, encoding for an ANL with a similar deletion, was identified (Appendix Fig. S3). We also screened other plant proteomes for homologs with similar features and identified several NLR proteins with a similar deletion and potential N-terminal N-myristoylation and S-acylation sites in *Brassica napus* (BnaCnng57760D), *Capsella rubella* (XP_006300767.1; CARUB_v10019842mg; Carub.0002s0399), *Capsella bursa-pastoris* (Cbp33710), *Cardamine hirsuta* (CARHR052110), *Leavenworthia alabamica* (LA_scaffold903_5), and *Cleome vicosa* (UPZX_scaffold_2005671). Phylogenetic analysis suggests that the deletion in the CC domain may have arisen at least twice independently (Appendix Fig. S4). Arabidopsis RNA-seq data indicate that *PML5* is expressed throughout the plant with the lowest expression in roots and the highest expression in rosette leaves (Klepikova et al, 2016; Zhang et al, 2020). These results demonstrate that ANLs with an N-terminal deletion and potential N-myristoylation and S-acylation sites are found throughout the Brassicales, suggesting a conserved and perhaps important function for PML5 and its paralogous ANLs.

## PML5 functions as a canonical cell death inducing ANL

Effector-dependent activation of CNLs often causes a strong cell death response (Chen et al, 2022; Chung et al, 2011; Li et al, 2019; Saur et al, 2019; Shao et al, 2003). NLR activation can be mimicked by introducing a single amino acid mutation in the conserved MHD motif of the NB-ARC domain, or, in some cases, by strong overexpression of wild-type NLRs (Adachi et al, 2023; Gao et al, 2011). To investigate whether PML5 can function as a cell-death

inducing ANL, we generated an autoactive MHD mutant version by substituting aspartate 362 for valine (D326V) and transiently overexpressed wild-type and the D362V PML5 proteins in *N. benthamiana*. Wild-type PML5 and the D362V mutant elicited strong cell death responses regardless of the promoter or the C-terminal tag used (Figs. 1C,D and EV1C,D). Mutation of the conserved lysine at position 74 to arginine (K74R) in PML5 is expected to inhibit the function of the P-loop motif, which is essential for nucleotide binding and thus NLR activity (El Kasmi et al, 2017; Tameling et al, 2002; Williams et al, 2011). Indeed, the introduction of the K74R mutation abolished PML5 cell death activity (Figs. 1C and EV3C,D). This was not due to reduced accumulation of the K74R mutant (Figs. 1C and EV1C,D). Together, these analyses show that PML5 functions as a canonical ANL, despite lacking most of the CC domain residues found in other CNLs.

The N-terminal myristoylation and S-acylation sites of the well-studied RPS5/PML7 are important for RPS5/PML7 immune functions (Qi et al, 2012). We wanted to learn whether mutation of the predicted N-myristoylation and S-acylation sites of PML5 also affects its ability to cause cell death, and found that the GCC(2,3,4)A and GCC(2,3,4)A,D362V mutants did not induce any cell death, in spite of being well expressed (Figs. 1C and EV1C,D), suggesting that N-myristoylation and S-acylation are required for PML5 cell death activity. Conserved hydrophobic residues in the alpha-1 helix of the CC domain have been shown to be important for cell death activity of many CNLs (Adachi et al, 2019; Wang et al, 2019). Amino acids 13 to 29 of the PML5 N-terminus are predicted to form an alpha helix that aligns with the amino acids of the alpha-1 helix of Arabidopsis ZAR1 and *N. benthamiana* NRC4 (Fig. EV1A,B). The alignment suggested that PML5 shares conserved hydrophobic residues at similar positions (Fig. EV1B). To test whether these residues in the putative alpha-helix of PML5 also contribute to PML5 cell death activity, we exchanged T15, L16, I19, F20, and L23, including the conserved hydrophobic residues, into alanine and glutamate (Fig. EV1B). As previously reported for AtZAR1 and NbNRC4, mutation of these residues to alanine or glutamate individually or in combinations of three completely abolished PML5 cell death activity (Fig. 1D) (Adachi et al, 2019; Wang et al, 2019b). Protein blots confirmed the expression of each single mutant and the T15A L16A F20A triple mutant. However, we had difficulties to detect the mCherry-tagged T15E L16E F20E

mutant in protein expression blots, possibly due to some instability but we were able to detect the Citrine-HA-tagged variant in our confocal microscopy analysis (see below). We conclude that like other CNLs, PML5 cell death activity requires these conserved (hydrophobic) residues in the N-terminal putative alpha-1 helix for proper cell death induction.

To determine whether PML5 overexpression also induces cell death in Arabidopsis in a P-loop-dependent manner, as it does in *N. benthamiana*, we generated stable transgenic *Col*-0 plants transformed with *UBIQUITIN10* promoter-driven estradiol-inducible *PML5-GFP* wild-type, MHD- and P-loop mutant constructs. Induced expression of wild-type PML5 and the D362V mutant, but not the K74R mutant, caused severe cell death responses in leaves of 4-week-old plants (Figs. 1E and EV1E). In addition, seedlings expressing wild-type PML5 or the D362V mutant were severely stunted and eventually died when germinated and grown on estradiol-containing medium (Fig. EV1F). Thus, PML5 over-expression initiates strong P-loop-dependent cell death responses in both *N. benthamiana* and Arabidopsis plants.

Activation of the Arabidopsis CNL and RNL ZAR1 and ADR1, respectively, leads to a measurable increase in cytosolic calcium concentration ($[Ca^{2+}]_{cyt}$), which is required for cell death induction (Bi et al, 2021; Jacob et al, 2021). This increase in $[Ca^{2+}]_{cyt}$ was dependent on their ability to form resistosomes and required the integrity of their alpha-1 helix. We tested whether expression of auto-activated PML5 in transgenic GCamp3 *N. benthamiana* (DeFalco et al, 2017) would lead to a similar increase in $[Ca^{2+}]_{cyt}$. Transient expression of PML5 D362V also induced a strong calcium influx similar to that of ADR1 D461V, and this increase in $[Ca^{2+}]_{cyt}$ was P-Loop-dependent (Fig. 1F). The general calcium channel inhibitor lanthanum chloride ($LaCl_3$) abolished cell death activity of PML5 in *N. benthamiana* leaves as well as the cell death activity of ADR1 (Fig. 1G (Jacob et al, 2021)). Thus, we conclude that PML5 may also form $Ca^{2+}$-permeable cation channels that mediate $[Ca^{2+}]_{cyt}$ increase to initiate a strong cell death response. Collectively, these data suggest that PML5, despite the deletion of 113 amino acids in its CC domain, can function as a canonical CNL to induce cell death and calcium influx when activated or transcriptionally induced. PML5 is thus a naturally occurring CNL with a minimal signaling domain and therefore makes for an excellent platform to understand the functional importance of this minimal domain, including its structure, for NLR signaling, NLR-mediated calcium influx and cell death induction.

## EDS1, RNLs, and NDR1 are not required for PML5 cell death activity

Many NLRs, including some effector sensing CNLs, require RNLs and the plant-specific lipase like proteins EDS1 (ENHANCED DISEASE SUSCEPTIBILITY 1), PAD4 (PHYTOALEXIN DEFI-CIENT 4), and SAG101 (SENESCENCE ASSOCIATED GENE 101) to induce cell death and immunity (Castel et al, 2019; Jubic et al, 2019; Lapin et al, 2020; Saile et al, 2020; Sun et al, 2021; Wu et al, 2019) To investigate whether PML5 may also require RNL or EDS1 function, we transiently expressed wild-type PML5 and the D362V mutant in *eds1*, *adr1*, *nrg1* single and *adr1 nrg1* double mutant *N. benthamiana* plants. In all mutant backgrounds, both wild-type PML5 and its MHD mutant induced cell death comparable to that in wild-type *N. benthamiana* leaves (Fig. EV2A). To determine

whether PML5 cell death activity in Arabidopsis is independent of RNLs and EDS1, we generated stable transgenic *helperless* (mutated for all five full-length RNL genes (Saile et al, 2020)) and *eds1-12* mutants conditionally expressing *PML5-GFP*. Induction of PML5-GFP expression in these mutants resulted in strong cell death responses (Fig. EV2B,C), suggesting that PML5 cell death activity is independent of RNL helper NLRs and EDS1 in both plant species. Some CNLs require the plasma membrane-localized protein NDR1 (NON-RACE-SPECIFIC DISEASE RESISTANCE 1) for cell death and resistance signaling (Century et al, 1995; Knepper et al, 2011; Shapiro and Zhang, 2001). To investigate whether PML5 cell death function might also require NDR1, we generated a stable transgenic *ndr1-1* mutant that conditionally expresses *PML5-GFP*. Again, expression of PML5 induced a strong cell death response similar to the expression of PML5 in the Col-0 background (Fig. EV1C). This demonstrated that PML5 induced cell death does not require NDR1. Taken together, these results suggest that PML5 functions as an autonomous NLR not requiring other known downstream immune components for cell death induction.

## The N-terminal 60 amino acids of PML5 are sufficient for cell death

Overexpression of the full-length CC domain of many CNLs and ANLs is sufficient to induce a strong cell death response, demonstrating that the N-terminal CC domain is the signaling, potentially pore-forming domain (Adachi et al, 2019; Baudin et al, 2020; Chia et al, 2024; Li Wan, 2019; Maekawa et al, 2011; Wang et al, 2020; Wroblewski et al, 2018). To determine whether the N-terminal partial CC domain of PML5 or another domain is sufficient for cell death induction, we generated different PML5 truncations (Fig. 2A). Based on the modeled AlphaFold 2 structure, we constructed three different CC-type fragments (CC$^{1-35}$, CC$^{1-39}$, and CC$^{1-60}$), three NB-ARC containing fragments (NB$^{36-363}$, NB$^{40-383}$, and NB$^{61-383}$) and two LRR fragments (LRR$^{364-762}$ and LRR$^{384-762}$) and expressed these variants in *N. benthamiana*. None of the constructs containing NB-ARC or LRR domain encoding fragments were able to induce visible cell death (Fig. 2B), but PML5 CC$^{1-60}$ consistently induced cell death, albeit weakly (Fig. 2B). Lack of cell death was not due to reduced expression, as confirmed by protein blots (Fig. 2B (lower panel)). These results demonstrate that the minimal PML5 CC domain is the cell death signaling domain.

To test whether different PML5 domains could complement each other in *trans*, as described for the *Solanaceae* CNL Rx (Peter Moffett, 2002), we co-expressed PML5 CC$^{1-60}$ with the corresponding NB-ARC$^{61-383}$ and LRR$^{384-762}$ fragments. Full cell death activity could not be achieved (Fig. EV3A), indicating that the first 60 amino acids of PML5 are capable of inducing cell death, but not to the same extent as the full-length protein.

Activation of and thus cell death induction by CNLs or their CC domains requires self-association or oligomerization (El Kasmi et al, 2017; Wang et al, 2019b). We sought to determine whether the cell death-inducing PML5 CC$^{1-60}$ fragment is also self-associating. Co-immunoprecipitation (Co-IP) of transiently expressed CC$^{1-60}$ revealed self-association of this 60 amino acids fragment (Figs. 2C and EV3B). We attempted to support the Co-IP results by bimolecular fluorescence complementation (BiFC). However, we did not observe a YFP complementation different to the negative controls for the PML5 CC$^{1-60}$ fragment (Fig. EV3C,D).

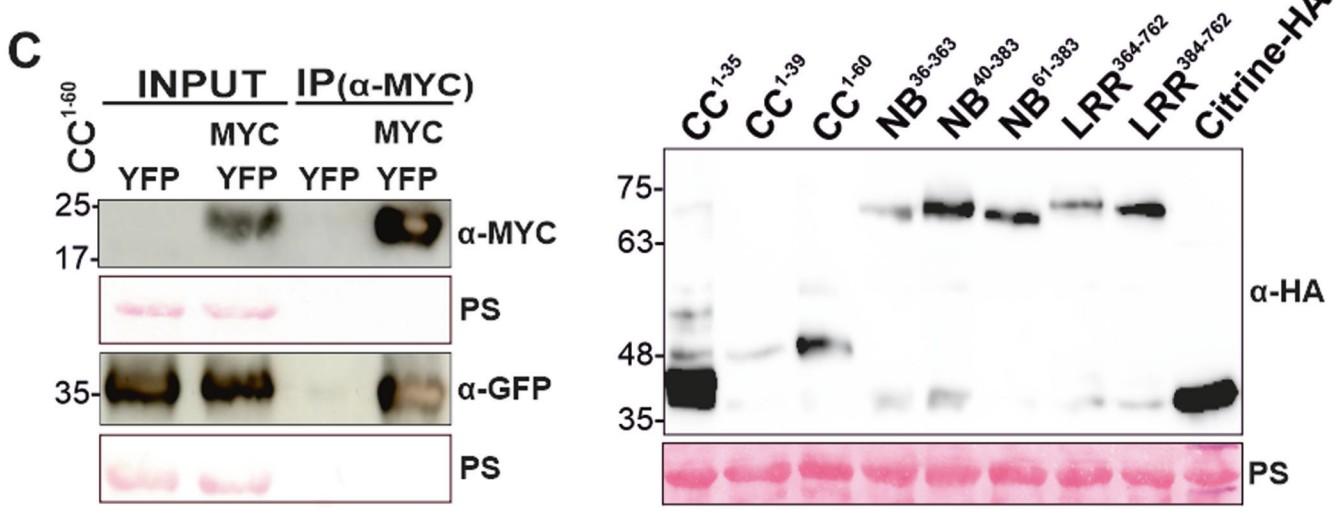

◄   **Figure 2.  PML5 N-terminal 60 amino acids self-associate and are sufficient for cell death induction.**

(**A**) Scheme of full-length PML5 and the generated PML5 fragments. Numbers indicate amino acid positions and length of fragments. (**B**) Cell death induced by transiently expressed full-length PML5 and PML5 fragments in *N. benthamiana* (top) and the corresponding protein blot (bottom). Leaves are shown in false color, red (top) or black/dark gray (bottom) indicates cell death and green (top) or light gray (bottom) healthy/alive tissue. Note: only full-length WT PML5 and the CC$^{1-60}$ fragment were inducing cell death symptoms. Photo was taken 2 days after induction with 20 µM estradiol. Fusion proteins were detected with an anti-HA (α-HA) antibody. Ponceau S (PS) staining is shown as loading control. Citrine-HA was used as a negative control. (**C**) PML5 CC$^{1-60}$ self-associates, as shown by transient expression in *N. benthamiana*. C-terminal Myc- and YFP-CC$^{1-60}$ fusions were co-expressed. Proteins were immunoprecipitated using anti-myc (α-MYC) beads and detected using anti-myc (α-MYC) and anti-GFP (α-GFP) antibodies. Ponceau S (PS) staining is shown as loading control. Source data are available online for this figure.

Taken together, the CC$^{1-60}$ fragment of PML5 is sufficient for cell death induction and self-associates, further confirming that PML5, despite lacking most of its CC$_{G10/GA}$ domain, induces cell death in a manner similar to other full-length CNLs.

## PML5 constitutively self-associates and forms higher-order complexes upon activation

We next examined whether the full-length wild-type and D362V mutant PML5 proteins self-associate. Co-IP experiments of differentially tagged PML5 indicated that not only the wild-type and D362V proteins self-associate (Fig. 3A,B), but also the non-cell death inducing K74R and GCC(2,3,4)A mutants (Fig. 3C,D). Attempts to support the Co-IP results by BiFC analysis failed. This may be due to steric hindrance of the fluorophore complementation when the PML5 proteins are C-terminally tagged.

Association of two proteins in Co-IP experiments suggests close spatial proximity of these proteins but does not necessarily demonstrate their ability to form oligomers, as hypothesized for activated CNLs. Blue native PAGE (BN-PAGE) is a suitable approach to study complex formation and thus oligomerization of proteins and has been successfully used to show CNL and RNL resistosome formation upon (auto-)activation (Ahn et al, 2023; Feehan et al, 2023; Hu et al, 2020; Jacob et al, 2021). To test whether PML5 forms resistosome-like complexes in an activation-dependent manner, we performed BN-PAGE with transiently expressed PML5 wild-type and mutants (Fig. 3E). The detection of protein complexes at ~550–600 kDa for all PML5 protein variants confirmed the Co-IP results, suggesting that regardless of activation status some PML5 proteins may form or be part of such complexes. Above 720 kDa and between 1048 kDa and 1236 kDa, we detected the presence of two PML5-containing complexes, specifically for PML5 wild-type and D362V. The potential PML5 complex above 720 kDa was also very weakly detected with the GCC(2,3,4)A and GCC(2,3,4)A, D362V mutants (Fig. 3E), indicating that the potential PTM sites at the N-terminus are not required for oligomerization of PML5. We did not observe the formation of the two complexes above 720 kDa for the K74R and the T15A,L16A,F20A loss-of-function mutants, suggesting that mutations within the P-loop and the conserved residues in the potential alpha-1 helix affect oligomerization, but not self-association. Thus, our Co-IP and BN-PAGE results indicated that only active PML5 forms a resistosome-like complex or at least associates with other proteins to form a high-molecular weight complex that can induce cell death.

## PML5 localizes to Golgi's and the tonoplast in a dynamic manner

Classification of PMLs based on their N-terminal N-myristoylation and/or S-acylation site(s) (Fig. 1A and Table EV1) suggests that

they are PM localized. Indeed, all PMLs characterized thus far have been reported to be PM associated (Gao et al, 2022; Huang et al, 2024; Huang et al, 2021; Qi et al, 2012; Yan et al, 2019; Zhang et al, 2017). Confocal laser scanning microscopy analysis of conditionally expressed fluorophore-tagged PML5 in *N. benthamiana* leaves revealed that PML5 wild-type exhibits a dynamic localization (Fig. 4). We observed PML5 colocalizing with a Golgi marker up to 6 h post induction (hpi) (Fig. 4A) and at the tonoplast (Fig. 4B,C) and cytosol 7 hpi and later (Fig. 4D). We did not detect any localization at the plasma membrane at the time points analyzed (Fig. 4E), which are just before cell death becomes macroscopically and microscopically visible. We confirmed membrane localization of wild-type PML5 by subcellular fractionation experiments (Fig. EV3E). At the time points when PML5 localized to tonoplast membranes, we observed a dramatic change in vacuolar morphology (Fig. 4B–E). All PML5 expressing cells showed this change in vacuolar morphology, a vesiculation or vacuolar fragmentation that was severely enhanced when the PML5 D362V mutant was expressed (Fig. EV3F). Stable transgenic Arabidopsis lines conditionally expressing PML5 showed a similar vacuolar fragmentation phenotype in elongating primary root cells (Fig. 4F) and revealed colocalization of PML5-GFP with the membrane dye FM4-64 at the tonoplast (Fig. 4G). These findings suggest a strong correlation between this vacuolar phenotype and the cell death activity of PML5 and further indicate that PML5 functions at the tonoplast to induce calcium influx and cell death.

The PML5 loss-of-function mutants K74R, GCC(2,3,4)A and GCC(2,3,4)A,D362V were predominantly localized to the cytosol (Fig. EV3G,H,J). We were unable to determine convincingly whether the GCC(2,3,4)A mutant proteins completely lost membrane association, because of the very close proximity of the cytosol and the tonoplast in fully developed and expanded leaf epidermal cells and the resolution limitations of confocal microscopy (Fig. EV3I). Colocalization analysis of the T15E,L16E,F20E (EEE) mutant with our Golgi marker indicates at least some colocalization for this loss-of-function PML5 mutant (Fig. EV3K), suggesting that this mutant is still membrane (Golgi and potentially tonoplast) associated. Most importantly, no vacuolar fragmentation phenotype was induced by any of the loss-of-function PML5 mutants (Fig. EV3G–K).

Our analyses suggest a dynamic localization of PML5, first being associated with Golgi membranes and then detectable at the tonoplast and in the cytosol. The Golgi and/or tonoplast localization may be required for the cell death activity, which is preceded by vacuolar fragmentation, whereas we think that the cytosolic localization may be due to the overexpression. Whether this vacuolar fragmentation is the cause, or the consequence of PML5-induced cell death is not yet clear. Thus, PML5 is the first ANL that has been shown to not localize to the

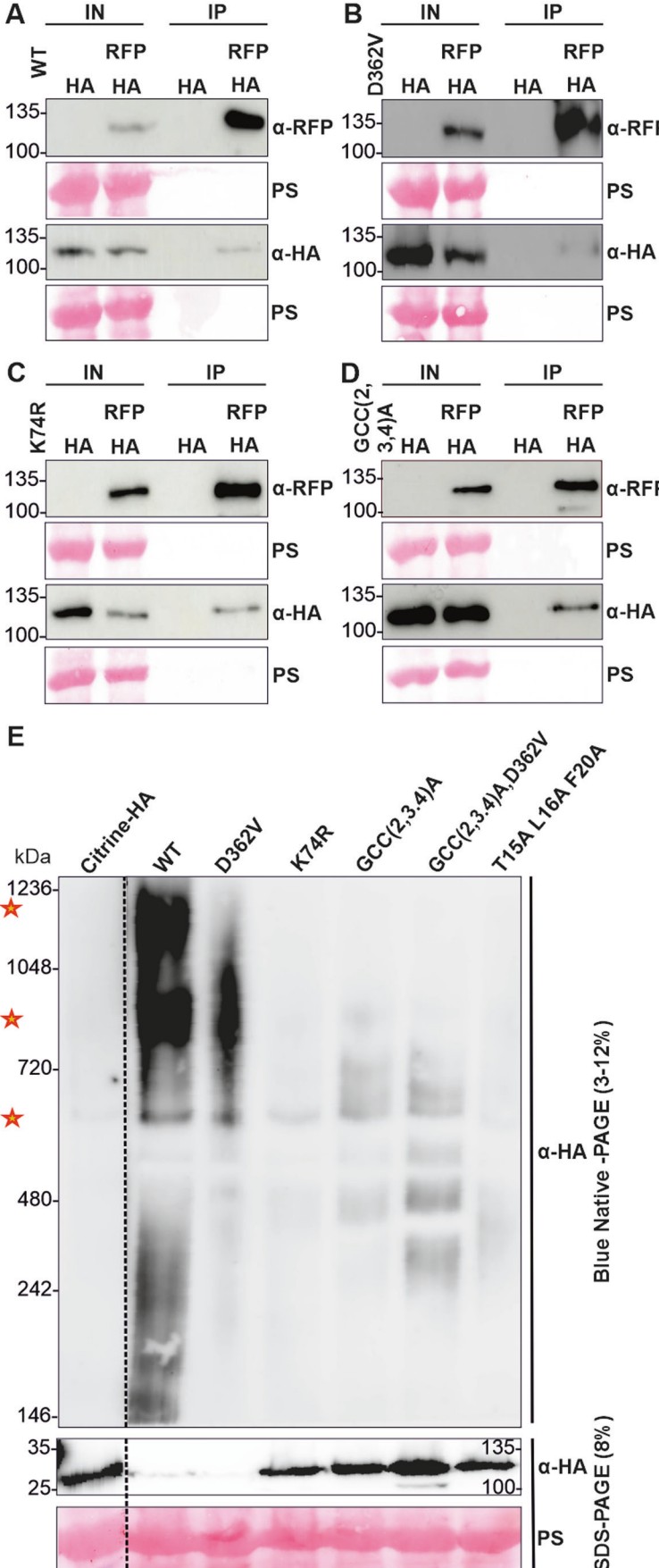

**Figure 3. Full-length PML5 self-associates and forms high-molecular weight complexes.**

(A–D) PML5 WT (**A**), D362V MHD mutant (**B**), K74R P-Loop mutant (**C**), and GCC(2,3,4)A mutant (**D**) self-associate, as seen after transient expression in *N. benthamiana*. Proteins were extracted 4 h after induction with 20 μM estradiol and immunoprecipitated using anti-RFP (α-RFP) beads and detected using anti-RFP (α-RFP) and anti-HA (α-HA) antibodies. Ponceau S (PS) staining is shown as loading control. (**E**) PML5 wild-type and D362V mutant form high-molecular weight complexes. Total protein from *N. benthamiana* leaves transiently expressing PML5 WT and PML5 mutants was extracted 6 h after induction with 20 μM estradiol, separated by BN-PAGE, and detected with an anti-HA (α-HA) antibody. Protein extracts were also separated by SDS-PAGE and proteins detected with anti-HA (α-HA) antibody. Ponceau S (PS) staining is shown as loading control. Red stars indicate potential PML5-containing complexes of different sizes. Source data are available online for this figure.

PM and to induce cell death, possibly by forming functional resistosomes at the tonoplast.

## Loss of *PML5* in Col-0 results in a modest increase in susceptibility to bacterial infection

As mentioned above, induced overexpression of active PML5 markedly induced cell death and altered vacuolar morphology (Figs. 1 and 4), suggesting that while PML5 functions as a canonical CNL it also exhibits distinct characteristics, such as dynamic localization to Golgi and tonoplast and a pronounced vacuolar fragmentation phenotype. To elucidate whether loss of *PML5* would have any effect on development or immunity, we used CRISPR/Cas9 gene editing to generate a knockout mutant, *pml5-2c*, with a 4192 bp deletion, spanning the entire coding sequence, which is replaced by a short 6 bp stuffer fragment (Fig. 5A and see Methods section). We characterized this mutant together with a *pml5* T-DNA insertion line, *pml5-1* (Salk_0691926) (Fig. 5A). The *pml5* mutants were morphologically indistinguishable from *Col*-0 wild-type (Fig. 5B,C) but flowered approximately four to seven days early in our green house and growth chambers (Fig. 5D). The *pml5-1* allele is not a complete knockout, as some residual transcript is generated from sequences upstream of the T-DNA insertion (Fig. 5E).

To learn whether *PML5* has a function during immunity induced upon infection with a bacterial pathogen, we infected the *pml5* mutants with the virulent *Pseudomonas syringae* pv. *tomato* (*Pst*) DC3000 strain. The two *pml5* alleles were slightly more susceptible to *Pst* DC3000 infections (Fig. 5F). This was also evident on the infected leaves, as disease symptoms were slightly more pronounced in *pml5-1* and *pml5-2c* compared to *Col*-0 wild-type plants (Fig. 5G).

To understand whether *PML5* is involved in ETI responses or in basal immunity, we did infections with *Pst* DC3000 expressing effectors that activate specific NLRs (RPM1, RPS5, ZAR1, and RPS4/RRS1) in *Col*-0 and the effector-depleted *Pst DC3000* D36E strain (Debener et al, 1991; Lewis et al, 2010; Narusaka et al, 2009; Simonich and lnnes, 1995; Wei et al, 2015). Sensor NLR-induced bacterial growth restriction and HR were not altered in *pml5-1* and *pml5-2c* mutants (Fig. EV4A–L). Similarly, we did not observe any effect on growth restriction of *Pst* D36E on *pml5* mutants (Fig. EV4M,N). The lack of a measurable effect on *Pst* D36E growth in *pml5* suggests that PML5 does not have an important function during PRR-mediated immunity. Indeed, reactive oxygen species (ROS) accumulation and ethylene production, two major PRR-induced responses, were not negatively affected upon treatment of *pml5-2c* with the PAMPs flg22 (triggers the PRR FLS2) (Mersmann et al, 2010) or nlp20 (triggers the PRR RLP23) (Bohm et al, 2014) (Fig. EV5A,B).

The immune response against the virulent bacterial pathogen *Pst* DC3000 was marginally affected by loss of PML5 (Fig. 5F). This could indicate a general function of PML5 in immunity that is not observed in resistance to avirulent or effector-less *Pst* DC3000 strains. To investigate this hypothesis further, we inoculated both *pml5* mutants and *Col*-0 with tobacco rattle virus (TRV) (Liu et al, 2002). TRV accumulation, as measured by quantification of viral RNA in the infected leaf tissue, was not significantly altered compared to wild-type (Fig. EV5C), indicating that PML5 does not contribute to anti-TRV resistance.

Collectively, these results suggest that PML5 may contribute to basal immunity against the virulent bacterial pathogen *Pst* DC3000 but does not contribute to resistance against the tested avirulent *Pst* DC3000 effector-containing strains and TRV.

## Discussion

We report the identification and initial characterization of a $CC_{G10/GA}$-type NLR (ANL) immune receptor with a conserved deletion of most of the N-terminal CC signaling domain. Our analyses demonstrate that despite its 113 amino acid deletion, PML5 functions as a canonical full-length ANL with respect to the initiation mechanism of cell death. PML5 has strong, P-loop-dependent cell death activity when overexpressed (Fig. 1), and residues 1 to 60 are sufficient for cell death induction (Fig. 2), suggesting that the N-terminal region of PML5 is the cell death executing part. This is surprising, because the PML5 N-terminus is most likely N-myristoylated and S-acylated, which may contribute to a constitutive membrane association of the PML5 reminiscent $CC_{G10/GA}$ domain. AlphaFold 2 structural prediction suggests that the PML5 $CC_{G10/GA}$ domain also lacks three of the four alpha helices normally found in full-length CNLs and ANLs (Bentham et al, 2018; Forderer et al, 2022). However, several hydrophobic residues, recently suggested to be required for proper function of all cell death inducing CNLs (Chia et al, 2024), are retained in the PML5 N-terminal signaling/executioner domain and are required for PML5 function (Fig. 1D). Thus, the remaining predicted alpha helix of PML5 may serve the same function as the alpha-1 helix of AtZAR1 or NbNRC4 (Adachi et al, 2019; Chia et al, 2024; Wang et al, 2019b). This is supported by our finding that PML5 expression also lead to a strong increase in $[Ca^{2+}]_{cyt}$, similar to the increase mediated by these CNLs and the Arabidopsis RNL ADR1 (Fig. 1E,F).

The potential N-myristoylation and S-acylation sites at the N-terminus of PML5 are shared by full-length members of the PML family and are important for at least RPS5 (Qi et al, 2012), R5L1 (Gao et al, 2022), L3 and L5 function (Huang et al, 2024; Huang et al, 2021). In our experiments, mutation of PML5 G2, C3, and C4

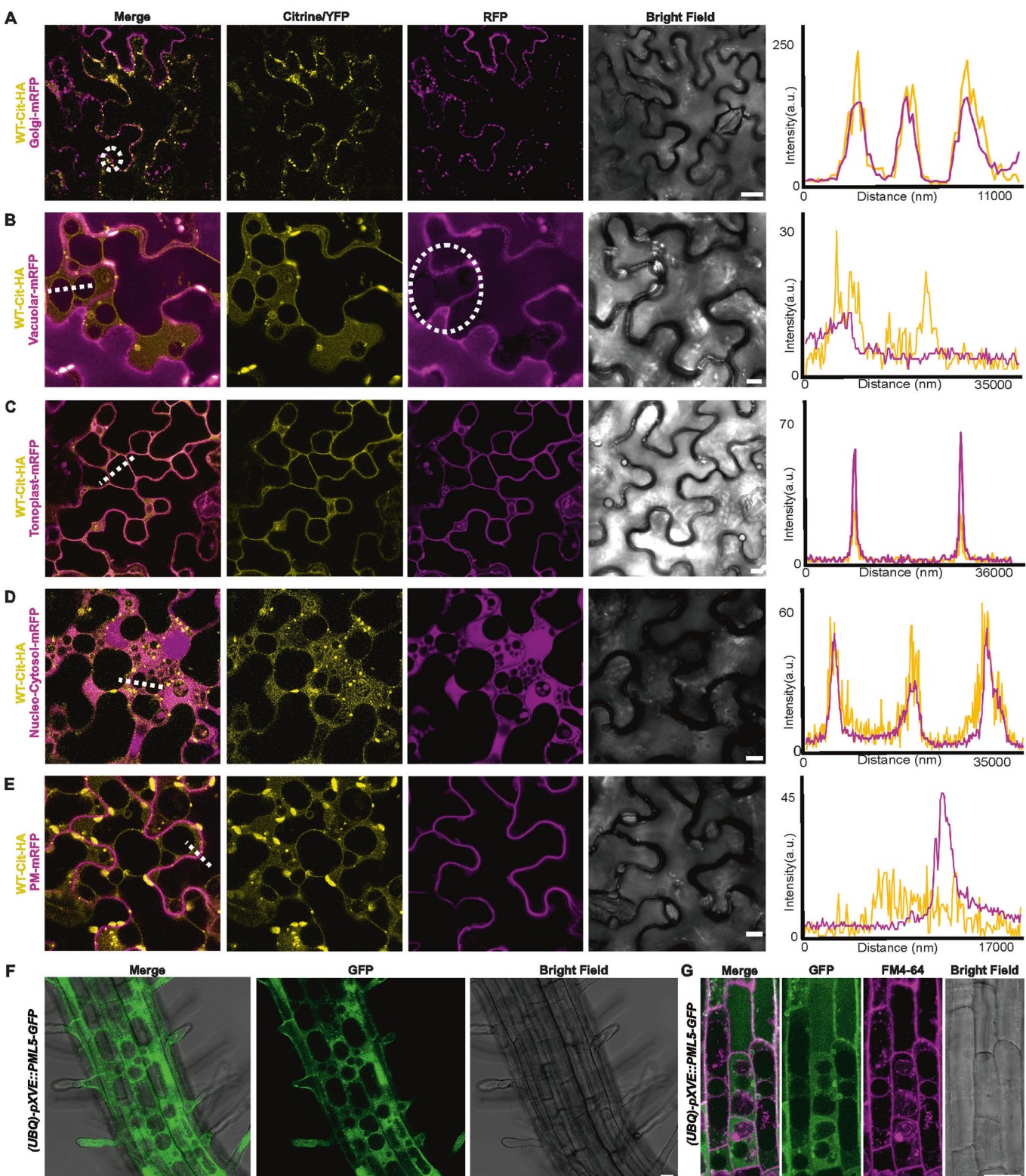

affected cell death activity and proper Golgi/tonoplast localization (Figs. 1C and EV3H,I,J), suggesting that PML5 activity also requires post-translational modifications of these residues. We hypothesize that pre-activation localization to the correct cellular membrane, such as the PM or other endomembrane compartments, is mediated by N-myristoylation and S-acylation of these residues and is contributing to proper activation and thus function of all PMLs. This is supported by the lack of cell death activity of the PML5 G,C,C(2,3,4)A (Fig. 1), RPS5 G(2,3)A,C4A (Qi et al, 2012), L3 and L5 G2A mutants (Huang et al, 2021 and 2024). Potentially,

Figure 4.  PML5 localizes to different cellular compartments.

(A–E) Confocal laser scanning microscopy indicates localization of transiently expressed PML5-Citrine-HA to Golgi membranes (A), the vacuolar membrane (B) and (C), and cytosol (D), but not to the plasma membrane (E). White dotted lines indicate the area used for colocalization profile analysis and the corresponding profiles are shown on the right, except for (A), where a line crossing the three circled puncta was used to extract the intensity profile. White dotted circle in (B) indicates vacuolar fragmentation. Scale bars, 10 μm; see Methods for compartment marker proteins. PML5-Citrine-HA expression was induced with 20 μM estradiol and confocal microscopy was performed 6–8 h after induction: Golgi (6 hpi), vacuolar membrane (7 hpi), cytosol (7 hpi), plasma membrane (8 hpi). (F) PML5-GFP localization in Arabidopsis elongating root cells of a transgenic line. Scale bar, 20 μm. PML5-GFP expression was induced with 20 μM estradiol and confocal microscopy was performed 18 h after induction. (G) PML5-GFP colocalizes with FM4-64 at the tonoplast. PML5-GFP expression was induced with 20 μM estradiol. Sixteen hours after induction seedlings were stained with 4 μM FM4-64 for 3 h before washed out. Scale bar, 20 μm. Confocal microscopy was performed 23 h after induction. Source data are available online for this figure.

S-acylation and N-myristoylation also enable or at least support oligomerization of activated PML5 (Fig. 3E). How such acylated PMLs are capable of inducing cell death by a mechanism similar to that proposed for CNLs or RNLs is still not fully understood. It is possible that after proper localization at the membrane and upon (effector-dependent) activation a de-palmitoylation of the modified cysteines or perhaps a phosphorylation of residues close to the PTM sites could release the very N-terminus from the inner leaflet of the membrane, allowing the full-length PMLs to oligomerize and penetrate the lipid bilayer with their alpha-1 helix for channel or pore formation enabling calcium influx and cell death initiation.

Cleavage of the very N-terminal region upon or prior to activation by an unknown protease may also be possible. However, we did not observe a size difference of cell death active wild-type or D364V mutant PML5 compared to the PML5 loss-of-function variants in our protein blots. Detection of such a very small N-terminal cleavage product would be difficult, because N-terminal tagging of CNLs negatively affects their cell death function and in the case of PMLs their S-acylation/N-myristoylation and thus their localization, which we and others have shown to be required for their activity (Wang et al, 2019b). Our results and previous published work on PMLs suggest that this family of CNLs most likely induces cell death by a very similar mechanism as CNLs with a so-called MADA, MADA-like, or MAEPL motif at their very N-terminus (Adachi et al, 2019; Chia et al, 2024). Oligomerization of activated CNLs and the RNL protein family leads to cation-permeable channel formation at the PM and $Ca^{2+}$ influx, and is required for cell death and immune function (Bi et al, 2021; Contreras et al, 2023; Forderer et al, 2022; Jacob et al, 2021; Wang et al, 2019b; Wang et al, 2019a). We could show that PML5 self-associates, most likely pre-activation, and that it oligomerizes upon activation (Fig. 3), potentially also forming resistosomes mediating $Ca^{2+}$ influx to induce cell death and resistance. CNLs, including all PMLs characterized so far, execute their cell death function at the PM (Gao et al, 2022; Huang et al, 2024; Huang et al, 2021; Qi et al, 2012; Yan et al, 2019; Zhang et al, 2017). However, PML5 is, to our knowledge, the first (Arabidopsis) CNL found to be Golgi and tonoplast localized. Our results strongly suggest that active PML5 exerts its cell death function at one of these compartments, most likely at the tonoplast. PML5 cell death activity was independent of PM-localized NDR1, which is required for the function of many PM-localized CNLs. This, together with the lack of PML5 PM localization, further supports the conclusion that PML5 does not function at the PM but rather at the tonoplast. PML5 may form a cation and $Ca^{2+}$ permeable channel at the tonoplast enabling the release of $Ca^{2+}$ from the vacuole into the cytoplasm. The vacuole, like the endoplasmic reticulum and the apoplast, is an important

storage of calcium in plants (Peiter, 2011; Schonknecht, 2013), and could very well be a source of $Ca^{2+}$ required for NLR-induced cell death. The interpretation that PML5 may release $Ca^{2+}$ from the vacuole into the cytoplasm is supported by the altered vacuolar morphology upon overexpression of cell death-active PML5. This vacuolar fragmentation/fission or tubulation leads to the appearance of bubble-like structures and in some cases also apparent membrane invaginations (Fig. 4), which is very similar to what was recently described for the overexpression of plant mechanosensors of the PIEZO protein family (Radin et al, 2021). There, tonoplast localized PIEZOs promote the release of $Ca^{2+}$ stored in vacuoles and modulate vacuolar complexity/morphology, allowing the plant cell to respond to changes in cell mechanics. Whether there is a relation between PML5-induced changes in vacuolar morphology and plant PIEZOs remains to be elucidated.

How did the deletion of most of the $CC_{G10/GA}$ domain occur independently at least twice during evolution, and why were PML5 homologs with such deletions not lost during evolution? At this point, we can only speculate about possible answers. PML5 and its three closest orthologs (PML11, 12, and 13) belong to the class of highly variable NLRs (hvNLRs) in Arabidopsis (Prigozhin and Krasileva, 2021). hvNLRs are specified by high allelic diversity, especially in the LRR coding region. It is conceivable that the nucleotide sequence at the 5' end (CC domain-encoding part) of these PMLs is also prone to deletions because of the presence of fragile DNA sites, similar to recurrent deletions that confer evolutionary advantages that have been found in other systems (Xie et al, 2019). This would be consistent with the finding that gene body mutations in Arabidopsis are more likely at the 5' and 3' ends of genes (Monroe et al, 2022). hvNLRs have been hypothesized to act as sources for novel recognition specificities of NLRs (Prigozhin and Krasileva, 2021), and it is likely that a specific effector (or pathogen) recognized by AtPML5 may not yet exist. In any case, the occurrence of PML5-like deletions in ANLs of other Arabidopsis accessions and Brassicales species indicates an evolutionary advantage of retaining this ANL variant in the population of diversified species. PML5-like NLRs might serve an important function in the immune system and are under balancing selection (Koenig et al, 2019). Balancing selection to maintain intra- and interspecific allelic diversity was also described for a barley NLR with a novel integrated Exo70 domain, which was also found in NLRs of unrelated *Poaceae* species that diversified from barley 24 million years ago (Brabham et al, 2018). Balancing selection has also been suggested for certain paired NLRs (Shimizu et al, 2022).

The exact function and contribution of PML5 and PML5-like NLRs in immunity requires further investigation. It would be interesting to see whether PML5-like ANLs of other plants are also executing their cell death function at the tonoplast. We observed a slight decrease in resistance to the hemibiotrophic pathogen *Pst*

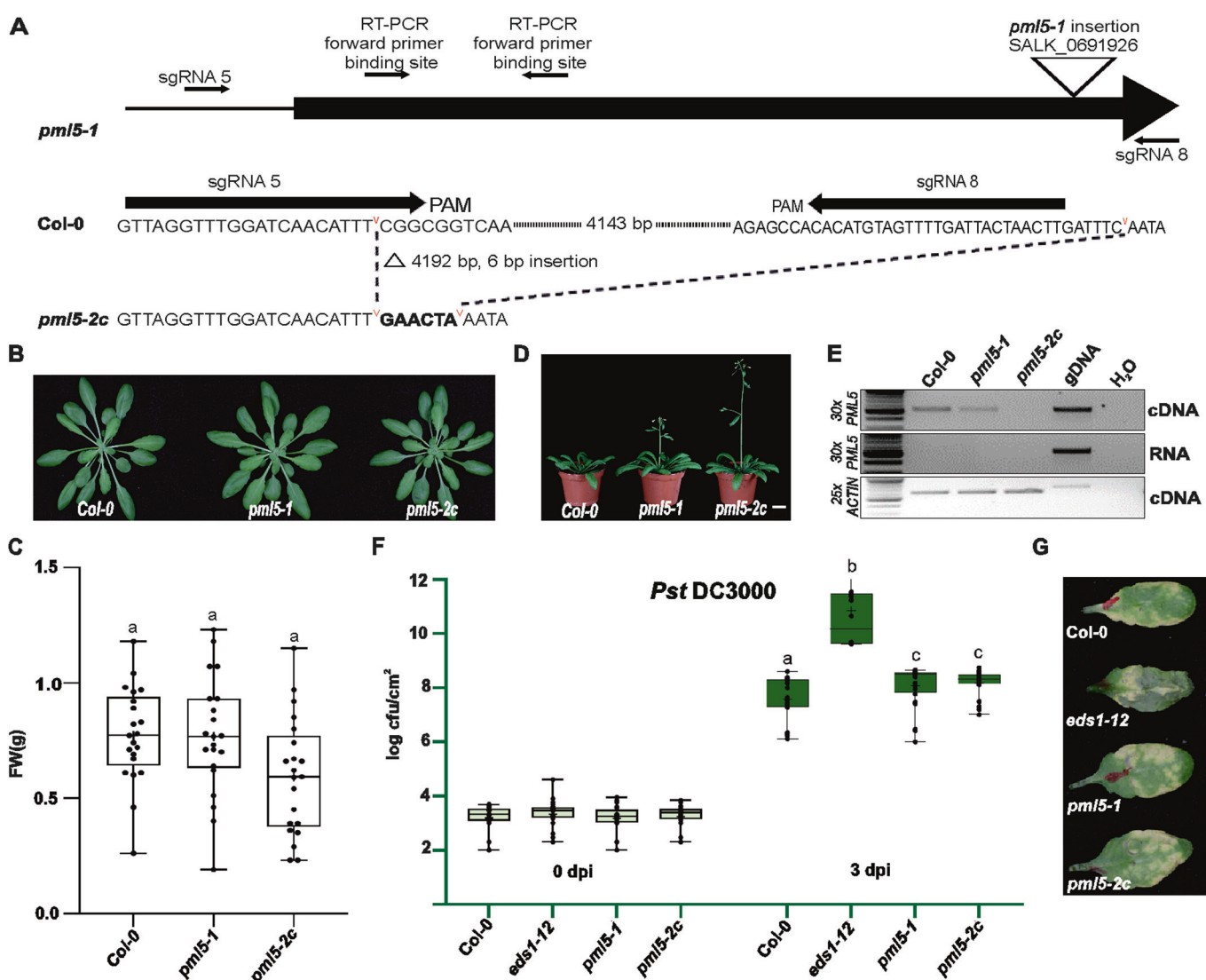

**Figure 5. *pml5* mutants are slightly more susceptible to *Pst* DC3000 infection.**

(A) Diagram of *pml5* alleles. SALK T-DNA insertion (*pml5-1*; *SALK_0691926*) and CRISPR/Cas9 induced 4192 bp deletion/6 bp insertion in *pml5-2c* are shown. (B) Rosette of 8-week-old Arabidopsis *pml5-1* and *pml5-2c* mutants compared to wild-type Col-0 grown in short day condition. (C) Fresh weight of 8-week-old wild-type, *pml5-1*, and *pml5-2c* plants from panel (B). Data is shown as boxplots. Data points represent 1 biological replicate with each 21 technical replicates (*n* = plants = 21). (D) Early flowering of 4-week-old *pml5-1* and *pml5-2c* mutants compared to same age Col-0 wild-type. Plants were grown in long day condition. Scale bars, 2 cm. (E) No detectable *PML5* transcripts in *pml5-2c* and reduced transcript levels in *pml5-1*, as determined by RT-PCR (30 cycles). *ACTIN2* (25 cycles) is used as positive control and genomic DNA (gDNA) (30 cycles) as negative control. (F) Six-week-old plants, hand infiltrated with *Pst* DC3000 ($OD_{600}$ = 0.001) support modestly enhanced bacterial growth in *pml5-1* and *pml5-2c*. Infiltrated leaves were collected on day 0 and day 3 post-infection for bacterial colony counts. Data are shown as boxplots and data points (colony forming units per square cm—cfu/cm²) represent two biological replicates with four technical replicates each (*n*8, each with six leaf samples). (G) *Pst* DC3000-induced disease symptoms on *pml5-1* and *pml5-2c*. Col-0 and the *eds1-12* mutants served as controls. Data information: For (C) and (F), individual data points are shown in boxplots (center line, median; bounds of box, the first and the third quartiles; whiskers, 1.5 times the interquartile range; error bar, minima and maxima. Data points with different letters indicate significant differences of $P \leq 0.05$ (one-way ANOVA with a post hoc Tukey's HSD test). Exact *P* values for all experiments are provided in Dataset EV1. Source data are available online for this figure.

DC3000 for *pml5* mutants under our growth conditions (Fig. 5). This has been reported previously for *pml5* and a number of other ANL mutants in Arabidopsis, including *rps5* and *r5l1* (Gao et al, 2022; Jiang et al, 2020). The transcription of many PMLs is regulated by the microRNA472-mediated silencing pathway and overexpression or knockout of miR472 decreases or enhances basal disease resistance against *Pst* DC3000, respectively (Boccara et al,

2014; Jiang et al, 2020). However, we would like to note that no transcriptional upregulation of *PML5* was reported during virulent or avirulent bacterial infections by RNAseq analysis (Saile et al, 2020; Yang et al, 2021). This is different for the Arabidopsis *R5L1*, which was found to be transcriptionally upregulated 48 h post *Pst* DC3000 infection (Gao et al, 2022). PMLs may function in or contribute to basal resistance in general, or they may have evolved

to recognize specific effectors or guard conserved effector targets, like for example RPS5, SUMM2, or SUT1. The tonoplast localized PML5, and potentially the truncated PML5-like ANLs in other species, may be required for mediating resistance against pathogens utilizing effectors that target and manipulate tonoplast-associated host proteins or serve as additional Ca²⁺ channels/pores at the tonoplast to boost calcium influx during specific immune responses, thereby contributing to full immunity.

## Methods

### Transgenes

The coding sequences of *PML5*, *eYFP-HA*, and *eYFP* were cloned into GATEWAY™ compatible vector pDONOR207 (Invitrogen, Carlsbad, USA) via GATEWAY™ Cloning Technology (Thermo Fischer Scientific, Waltham, USA) using primers listed in Table EV2. All mutant constructs were generated by Site-Directed mutagenesis PCR using primers listed in Table EV2. Two individual PCR reactions were prepared using either the reverse or forward primer containing the mutation of interest and incubated in the appropriate PCR conditions. 10 μl of the PCR products were digested overnight with *DpnI* (NEB, Frankfurt am Main, Germany) and transformed into *Escherichia coli* DH5α. All constructs were confirmed by Sanger sequencing (Eurofins, Ebersberg, Germany). LR reactions (Gateway Cloning™ Technology Thermo Fischer Scientific, Waltham, USA) were performed to introduce the CDS into a modified 35S promoter-driven estradiol-inducible destination vector pMDC7-Citrine-HA (Curtis and Grossniklaus, 2003), Ubiquitin promoter-driven estradiol-inducible destination vector pMDC7-GFP, estradiol-inducible destination vector pABindmCherry (Bleckmann et al, 2010), 35S promoter-driven destination vector pGWB641, pGWB617 (Nakamura et al, 2010) and pBAT-N, pBAT-C, pCL112, and pCL113 BIFC vectors (Urano et al, 2012). The different domain constructs for PML5 were amplified from full-length CDS with primers listed in Table EV2. Gateway Cloning™ was performed as described above.

### Plant material and growth conditions

*N. benthamiana* plants used in the study were grown in pots with soil for 4–5 weeks in a walk-in growth chamber under the following conditions: 12-h light/12-h dark, 24 °C/22 °C, with relative humidity (RH) of up to 70%. *N. benthamiana* mutant *eds1* (Schultink et al, 2017), *nrg1* (Qi et al, 2018), *adr1*, *adr1 nrg1* were obtained from Jeff Dangl's lab, the transgenic GCamp3 (DeFalco et al, 2017) lines from Keiko Yoshioka's lab. *A. thaliana* plants were grown in walk-in growth chambers with short day (8-h light/16-h dark in 21 °C/18 °C, 45% RH) or long day conditions (16-h light/8-h dark in 21 °C/18 °C, 45% RH). *A. thaliana eds1-12* (Ordon et al, 2017) and *ndr1-1* (Century et al, 1995) mutants were from the El Kasmi lab stock.

### Transgenic plants

*A. thaliana* Col-0 and *helperless* plants at the flowering stage were dipped with *Agrobacterium tumefaciens* strain GV3101/pMP90 carrying the specific constructs (Clough and Bent, 1998). Transformants were selected on ½ Murashige and Skoog (MS)

agar plates containing 15 μg/ml Hygromycin (Invitrogen, Thermo Fisher Scientific, Carlsbad, CA). Transgene expression was checked by immunoblotting total protein extracts with α-GFP antibodies. PML5-GFP expressing *eds1-12* and *ndr1-1* mutants were generated by crossing *PML5-GFP Col-0* with the specific homozygous mutant and homozygosity for the transgene in F2 and F3 generation was determined by hygromycin segregation and for the mutant allele by PCR genotyping.

### CRISPR/Cas9 genome editing

The sgRNA sequences unique to the *PML5* locus with predicted high efficiency were selected using CCTOP (Labuhn et al, 2018; Stemmer et al, 2015). Dicot genome editing vectors (pDGE) were used as the cloning system (Grutzner et al, 2021; Ordon et al, 2017; Stuttmann et al, 2021). Oligonucleotides containing the *PML5*-specific sgRNA sequences and overhangs for BpiI cut ligation into shuttle vectors pDGE332, pDGE333, pDGE335, and pDGE337 were ordered from Merck (Darmstadt, Germany). The assembled shuttle vectors were handled according to the polyclonal approach recommended in the provided instructions (Stuttmann et al, 2021) and used for BsaI cut ligation into the binary vector pDGE347. To confirm the correct sgRNA sequences, the final vector was sequenced with the recommended primers and transformed into *Agrobacterium tumefacien*s for *Arabidopsis* floral dip transformation (Clough and Bent, 1998). Transgenic T₁ seeds were selected using a stereoscope with RFP filter. Transgenic T₁ plants were genotyped with primers for deletion of the *PML5* locus (FEK_1215/FEK_1218), positive PCR products were Sanger sequenced (Eurofins, Ebersberg, Germany) after gel extraction of the band. In T₂, non-red seeds were selected, and the plants grown were genotyped again for *pml5* deletion and absence of the pDGE347 transgene (FEK_989/FEK_1095). In T₃ and T₄ generations, genotyping was repeated to confirm the results.

### T-DNA line

SALK_0691926 T-DNA line for *At1g61300/PML5* was obtained from the *Nottingham Arabidopsis seeds centre* (*NASC*). The insertion site was confirmed by sanger sequencing and is at 2289 bp of the CDS.

### RT-PCR

50 μg of 6-week-old *A. thaliana* leaf tissue were collected in a 2 mL reaction tube, frozen in liquid N₂ and crushed in a bead mill using a stainless-steel bead. Total RNA was extracted using the RNeasy kit (Qiagen). RNA concentration was determined, and equal amounts were used to perform a DNAse I (Thermo Scientific) digest for 30 min at 37 °C. Again, RNA concentration was determined, and equal amounts were used for RT-PCR with Superscript II reverse transcriptase (Thermo Fisher Scientific, Waltham, USA) according to the manufacturer's protocol. 2 μl of RNA or cDNA were used for PCRs. In addition, 2 μl of genomic DNA extracted with SENT buffer (Edwards et al, 1991) was used as a control experiment.

### Biomass quantification

Eight-week-old Arabidopsis plants grown under short day conditions were used. Shoot tissue was cut and weighed. Graphs were plotted using GraphPad Prism 9.

## Transient expression in *N. benthamiana*

*Agrobacterium tumefaciens* overnight cultures were centrifuged and resuspended in induction buffer (10 mM MgCl$_2$, 10 mM MES pH 5.6, 150 µM acetosyringone). The optical density at 600 nm (OD$_{600}$) of all constructs was adjusted to 0.3 and for *35S::P19* to 0.05. *35S::P19* was mixed with all the constructs to be infiltrated, to avoid any possible silencing issues. *Agrobacterium* mixtures were hand-infiltrated into the abaxial site of leaves of 4–5-week-old *N. benthamiana* plants. *Citrine-HA* or *35S::GFP* single infiltrations were done as control infiltrations for the experiments. Protein expression was induced 20 h post-infiltration by spraying using indicated estradiol (Sigma-Aldrich) concentration and 0.001% (v/v) Silwet L-77.

## Cell death assay

*Nicotiana benthamiana* leaves infiltrated with *Agrobacterium tumefaciens* cultures containing the specific constructs were imaged for cell death at indicated time points. Cell death images were taken under UV light using an Amersham ImageQuant 800 and integrated Cy3 and Cy5 filters (GE Healthcare, Chalfont St. Giles, UK).

Arabidopsis seedlings were sown on ½ MS agar plates supplemented with 20 µM estradiol and grown for 10 days under long-day conditions (16-h light/8-h dark in 21 °C/18 °C, 45% humidity). Growth defects were photographed using a Canon EOS 80D DSLR camera.

## In planta semi-quantification of Ca$^{2+}$ influx

Four- to six-week-old transgenic *GCamp3 N. benthamiana* leaves were infiltrated with *Agrobacterium tumefaciens*. Sixteen hours after infiltration, leaf discs (0.5 cm diameter) were harvested and placed abaxial side up in the wells of a microplate (MICROPLATE, 96 WELL, F-BOTTOM (Greiner BIO-ONE) filled with 100 µl MQ water and equilibrated for 4–6 h in the plant growth chamber. After equilibration for 4 h, expression was induced by adding 100 µl 80 µM Estradiol + 0.002% Silwet L-70. GCamp3 fluorescence was measured for 20 h using a BertholdTech Tristar$^2$ Multimode reader with the following settings: Counting time = 0.1 s, Exc filter 485/14, Em filter 535/25, 100% Xenon lamp energy, photomultiplier tube current: 750 V, sensitivity: medium. Analysis was done in R (Studio). Plotted values are averages of 24 leaf discs from 6 leaves of individual plants, with Standard Error represented. Western blots of additionally induced leaf samples collected 4 hpi were performed to show expression of mCherry-tagged fusion proteins.

## Lanthanum (LaCl$_3$) treatment

Cell death inhibition by LaCl$_3$ treatment was performed by adding LaCl$_3$ to the induction buffer prior to infiltration to a final concentration of 2 mM. The OD$_{600}$ for the *ADR1* and *PML5* constructs was set to 0.8 and for P19 to 0.2. Expression of estradiol-inducible constructs was induced 20 h after infiltration by spraying 100 µM estradiol + 0.002% Silwet ® L-77. Samples for protein extraction and western blot were collected 5–6 h later. Leaves with cell death phenotypes were photographed 24 h after induction of expression.

## Immunoblotting

For total protein extraction, leaf tissue from *N. benthamiana* or Arabidopsis was ground using a tissue lyzer (Mill Retsch MM400, Retsch GmbH, Haan, Germany) and resuspended in 190 µl ice-cold grinding buffer (20 mM Tris-HCl pH-7, 150 mM NaCl, 1 mM EDTA pH-8, 1% (v/v) Triton X-100, 0.1% (w/v) SDS, 5 mM DTT, 1X Halt™ PIC (Thermo Fisher Scientific, Waltham, USA). Samples were incubated on ice for 10 min and then centrifuged for 15 min at 16,000 × g and 4 °C. Then 5X SDS loading buffer (250 mM Tris-HCl pH 6.8. 50% (v/v) glycerol, 500 mM DTT, 10% (w/v) SDS, 0.005% (w/v) bromophenol blue) was added to the supernatant. Proteins were denatured at 95 °C for 5 min. Proteins were separated into 8% or 10% sodium dodecyl sulfate (SDS) polyacrylamide gels. The proteins were transferred to Amersham™ Protran™ 0.45 µm Nitrocellulose 300 mm membranes for western blotting. Primary and secondary antibody dilutions used for immunodetection were as follows: α-HA(rat)-1:2000 (clone 3F10, #11867423001; Roche Diagnostics, Basel, Switzerland), α-GFP(mouse)-1:1500 (clones 7.1 and 13.1, #11814460001; Roche Diagnostics, Basel, Switzerland), α-RFP(rat)-1:1000 (clone 5F8, #5f8; ChromoTek, Planegg-Martinsried, Germany), α-Myc(rat)-1:1000 (clone 9E1, #9e1; ChromoTek, Planegg-Martinsried, Germany), α-UGPase(rabbit)-1:2000 (#AS05086; Agrisera, Vännäs, Sweden), α-Histone H3(rabbit)-1:5000 (#AS10710; Agrisera, Vännäs, Sweden), α-mouse HRP-conjugated 1:10,000 (#A2554; Sigma-Aldrich, St. Louis, USA), α-rat HRP-conjugated 1:10,000 (#31470; Thermo Fisher Scientific, Waltham, USA), α-rabbit HRP-conjugated 1:10,000 (#A6154; Sigma-Aldrich, St. Louis, USA). Chemiluminescence was detected using an Amersham Image-Quant 800 (GE Healthcare, Chalfont St. Giles, UK). Images were processed with Coral Photo Paint (Corel Corporation, Ottawa, Canada) to adjust brightness and contrast.

## Co-immunoprecipitation

*Nicotiana benthamiana* leaf samples were frozen and ground in liquid nitrogen. Tissue powder was resuspended in 2.5 ml extraction buffer (50 mM HEPES buffer pH 7.5, 50 mM NaCl, 10 mM EDTA pH 8.0, 0.5% [v/v] Triton X-100, 5 mM DTT, 1x Halt™ Protease Inhibitor Cocktail (Thermo Fisher Scientific, Waltham, USA)). Samples were kept on ice for 20 min and centrifuged at 16,000 × g for 15 min. The supernatant was subjected to immunoprecipitation for 1 h using anti-GFP (ChromoTek, Planegg-Martinsried, Germany), anti-RFP (ChromoTek, Planegg-Martinsried, Germany) or anti-Myc beads (ChromoTek, Planegg-Martinsried, Germany). Beads were collected by centrifugation at 2400 × g at 4 °C and washed two times with 1 ml wash buffer (50 mM HEPES buffer pH-7.5, 150 mM NaCl, 10 mM EDTA pH-8.0, 0.2% [v/v] Triton X-100, 5 mM DTT, 1x Halt™ Protease Inhibitor Cocktail (Thermo Fisher Scientific, Waltham, USA)) by incubating the extracts for 5 min on a rotating wheel at 4 °C and two additional times by inverting the tube six times. Bound proteins were eluted in 120 µl 2x SDS loading buffer (100 mM Tris-HCl pH 6.8, 20% [v/v] glycerol, 200 mM DTT, 4% [w/v] SDS, 0.002% [w/v] bromophenol blue) and denatured by boiling the proteins at 95 °C for 5 min.

## Microsomal fractionation

Microsomal membrane fractions were prepared from 100 to 200 mg *N. benthamiana* leave tissue transiently expressing the indicated construct. Plant tissue as ground in liquid nitrogen and 2 ml sucrose buffer (20 mM Tris pH 8.0, 0.33 M sucrose, 1 mM EDTA, 5 mM DTT, and 1X Halt™ PIC (Thermo Fisher Scientific)) was added. Samples were centrifuged at $2000 \times g$ for 10 min and the supernatant were used for the following fractionation. 120 µl were taken away and used as the total protein extract sample (T) and the remaining supernatant was subjected to centrifugation at $17,000 \times g$ for 1 h at 4 °C. The pellet obtained was resuspended in 50 µl sucrose buffer and represents the microsomal/membrane fraction (M). The supernatant is the cytosolic fraction (S). The fractions obtained were used for SDS-PAGE.

## Blue native PAGE (BN-PAGE)

*Nicotiana benthamiana* leaf discs ($n = \sim 13$ (4 mg per disc), 50 mg) were ground in microcentrifuge tubes using a tissue lyzer (Mill Retsch MM400, Retsch GmbH, Haan, Germany) and mixed with ice-cold modified GTEN-DDM buffer (3.6 µl/mg) (10% glycerol, 100 mM Tris-HCl, pH 7.5, 1 mM EDTA, 150 mM NaCl, 5 mM dithiothreitol (DTT), 1X plant protease inhibitor cocktail (Halt™ PIC (Thermo Fisher Scientific), 0.5% (w/v) DDM (n-dodecyl β-D-maltoside) (Avanti Polar Lipids, Inc, USA)). The mixture was vortexed and placed on ice, then centrifuged at $20,000 \times g$ in 4 °C. The supernatant is the BN-PAGE sample. This supernatant was mixed with 4X native PAGE sample buffer (Invitrogen, Thermo Fisher Scientific, Carlsbad, CA) in the ratio of 3:1. Further add 0.5 µl of NativePAGE 5% G250 sample additive (Invitrogen, Thermo Fisher Scientific, Carlsbad, CA) to the 19.5 µl of the previous mixture and mix it well. This processed sample was used for running the BN-PAGE gels. Immunoblotting and detection of the complexes were carried out using the methods described in (Jacob et al, 2021).

## Confocal laser scanning microscopy

Confocal laser scanning microscopy was performed using a Zeiss (Oberkochen, Germany) LSM 880 laser-scanning confocal microscope equipped with a 40× water immersion objective. Images were obtained and analyzed using Zeiss ZEN blue or ImageJ (Schindelin et al, 2012) softwares for adjusting the brightness and contrast. Citrine and YFP fluorescence were excited at 514 nm, GFP at 488 nm, and RFP at 561 nm and emission was detected at 510–570 nm, 516/519–550 nm, and 588–651 nm, respectively. Chlorophyll A was excited at 561 nm and detected at 656–676 nm. The constructs for the cellular markers used in the study are *p35S::CD3-967-mCHERRY* as Golgi marker (Nelson et al, 2007), *p35S::sp-RFP-AFNY-STOP-tRFP* as a vacuolar lumen marker, *pUBQ::RFP* as a cytosolic marker, and *p35S::BRI1-RFP* as plasma membrane marker. To generate a tonoplast marker, we cloned the N-terminal 22 amino acids of Arabidopsis CBL6 (AT4g16350) as a level I module for Golden Gate cloning (Binder et al, 2014), with B-C overhangs. After a BsaI cut ligation, the assembled level II *p35S::CBL6-mCherry* construct was used for co-infiltration and colocalization studies. Confocal microscopy with transgenic *PML5-GFP* Arabidopsis seedlings was done with 7-day-old seedlings grown on ½ MS solid media under long-day conditions in a growth cabinet. Seedlings were placed in liquid ½ MS media supplemented with 20 µM estradiol 23 h before imaging. Sixteen hours after induction, the membrane staining dye FM4-64™ (Invitrogen, Thermo Fisher Scientific, Carlsbad, CA) was added at a final concentration of 4 µM. Seedlings were placed in fresh ½ MS liquid media about 3 h post FM4-64 treatment to wash excess FM4-64 away and imaging was done 3 to 5 h later to allow the uptake of FM4-64 into the tonoplast. PML5-GFP and FM4-64 were excited at 488 nm and at 561 nm, respectively, and emission was detected at 490–553 nm, and 570–745 nm, respectively.

## Bimolecular fluorescence complementation (BiFC)

For BiFC, *Agrobacterium tumefaciens* ($OD_{600} = 0.2$) containing the indicated constructs were mixed in a 1:1 ratio, with P19 expressing *A. tumefaciens* ($OD_{600} = 0.05$), and hand-infiltrated into leaves of 4-week-old *N. benthamiana*. YFP fluorescence was imaged at indicated time points using confocal laser scanning microscopy.

## Bacterial infection assay

*Pseudomonas syringae* strains used in the study were grown on KB agar plates with relevant antibiotics. Overnight grown bacteria were resuspended in 10 mM $MgCl_2$ and diluted to the required final concentration. Bacterial culture was hand infiltrated into rosette leaves of 6-week-old plants grown under short-day conditions. Leaf disc samples were collected on day 0 (0 dpi) and day 3 (3 dpi), ground in 200 µl distilled water using a tissue lyzer (Mill Retsch MM400, Retsch GmbH, Haan, Germany). Dilution series of the bacterial culture were plated on KB agar plates with appropriate antibiotics and cycloheximide, and colonies were counted 2 days later after incubation at 28 °C. Replicates were analyzed by one-way ANOVA with post hoc Tukey's test using GraphPad Prism 9. Bacterial cultures for cell death assays were hand-infiltrated on the right side of the leaf. Leaves were removed at the indicated time points, and autofluorescence was measured using a Typhoon FLA9500 laser scanner (Cytiva, Chicago, IL) of the adaxial side of the leaf. Image processing was done using ImageJ.

## Cell death quantification in Arabidopsis leaves

Autofluorescence of infiltrated tissue was quantified in ImageJ (Schindelin et al, 2012) by measuring the mean gray value of the pixels within a circular selection of either 28 pixels or 30 pixels radius depending on the smallest infiltrated area of the replicate. To offset background fluorescence, each measurement was divided by the average pixel intensity of each replicate calculated from measurements performed on non-infiltrated areas of the leaves as described above. Significance was evaluated on means minus the average background fluorescence of the leaves of the replicate using Tukey's HSD ($\alpha = 0.05$).

## Virus infection assay

Tobacco rattle virus (Liu et al, 2002) was used. *Agrobacterium* containing pYL156-TRV1 or pYL156-TRV2-EV plasmids were cultured in LB medium (containing 100 mg/L Kan and 25 mg/L Rif) for 18 h at 28 °C by shaking at 220 rpm. *Agrobacterium* cells were

then collected by centrifugation at $4000 \times g$ for 15 min, followed by resuspension in resuspension buffer (10 mM $MgCl_2$, 100 µM acetosyringone, and 10 mM MES, pH 5.8) and incubated at 28 °C for 3 h in the dark before infiltration into *N. benthamiana* plants. After 2 weeks, virus-infected leaves were harvested and ground on ice with extraction buffer (10 mM $Na_2HPO_4/NaH_2PO_4$, pH 7.2). The extracts were filtered through Miracloth and centrifuged at 4 °C for 1 min at $12,000 \times g$. Subsequently, the 4-week-old Arabidopsis plants were inoculated directly by rubbing the fresh supernatant of extracts generated from 2-week-old TRV-infected *N. benthamiana* plants onto rosette leaves. The Arabidopsis leaves were harvested at 6 and 8 days after inoculation, total RNA was extracted, and viral RNA loads were analyzed by qRT-PCR using Primer pairs (TRV-F: GTGCACGCAACAGTTCTAATCG and TRV-R: GCTGTGCTTTGATTTCTCCACC; Actin-F: CAGT GTCTGGATCGGTGGTT and Actin-R: TGAACGATTCCTG GACCTGC). The experiments were repeated three times with similar results.

## PTI experiments (ROS and ethylene measurements)

Leaves of 5–6-week-old Arabidopsis were cut into pieces of equal size and floated on $H_2O$ overnight. For ROS burst, the leaf pieces were transferred to a 96-well plate (1 piece/well) containing 100 µl of solution with 20 µM L-012 (Waco) and 2 µg $ml^{-1}$ peroxidase. Luminescence after treatment with peptides or water (as control) was measured with a luminometer (Mithras LB 940, Berthold) in 2 min intervals. Total relative light unit (RLU) production was determined by calculating the area under the scatter curve for the time points indicated. The RLU values at time point 0 min were set as 0 to accordingly remove the background responses. For ethylene production, three leaf pieces were incubated in a sealed 6.5 ml glass tube with 0.4 ml 20 mM MES buffer, pH 5.7 and the indicated elicitors. Ethylene production was measured by gas chromatographic analysis (GC-14A; Shimadzu) of 1 ml air from the closed tube after incubation for 4 h.

## In silico analysis

Phylogenetic trees and alignments in Figs. EV1, 2, 4 and 5 were constructed using CLC Main Workbench 23.0.1 (Qiagen, Venlo, Netherlands). The structure prediction for PML5 was done using AlphaFold 2 (https://alphafold.ebi.ac.uk/) and the UniProt ID of PML5 is O64790 DRL17_ARATH.

To examine whether the PML5 deletion in Arabidopsis is unique in the Brassicales we constructed a phylogeny using PML5 and homologs without the deletion from other representative Brassicalee genomes. Using Orthofinder (Emms and Kelly, 2019) we clustered the predicted protein sequences from three Capsella genomes (*Capsella bursa pastoris*, *Capsella rubella*, and *Capsella orientalis*, unpublished) and 18 Arabidopsis genomes. Using mafft (Katoh and Standley, 2013) we then aligned the resulting orthogroup containing the PML5 homologs with additional homologs from existing proteomes (*Brassica napus* (BnaCnng57760D), *Capsella rubella* (XP_006300767.1, Car-ub.0002s0399), *Capsella bursa pastoris* (Cbp33710), *Cardamine hirsute* (CARHR052110), *Leavenworthia alabamica* (LA_scaf-fold903_5). We then trimmed the amino acid alignment to just the NB-ARC region, which does not contain the deletion, and

constructed a maximum-likelihood phylogeny using model selection in IQtree that selected the JTT model (model JTT, ultra-fast bootstraps: 1000) (Nguyen et al, 2015).

## Funding

## Data availability

No primary datasets have been generated and deposited.

The source data of this paper are collected in the following database record: biostudies:S-SCDT-10_1038-S44319-024-00240-4.

## Peer review information

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

## Acknowledgements

We thank Bettina Hause for the CD3-967 construct, Manoj K Singh (Gerd Jürgens Lab) for the (UBQ)-pXVE-GFP empty vector and Klaus Harter for the BRI1 construct, Elke Sauberzweig, Christel Kulibaba-Mattern and Patrick Vetter for technical support, the ZMBP gardeners and microscopy facility for their

support and advice, Xander Zuijdgeest, Frank Vogt and Svenja Saile for critical reading of the manuscript, and other members of the El Kasmi, Nishimura and Dangl labs as well as Sandra Richter and Paul Gouguet for critical comments and discussions. We appreciate the support of Prabha Manishankar (Oecking Lab) and Nak Hyun Kim (Dangl Lab) for helping us to set up the BN-PAGE assay. This work was supported by a grant from the German Research Foundation (DFG) to FEK: (DFG EL 743/3-1) and the core funding from the University of Tübingen to FEK. NLR research in the El Kasmi, Nürnberger, Lozano-Duran and Weigel labs is also supported by DFG Grant CRC1101. Research in RLD's lab is partially funded by the Excellence Strategy of the German Federal and State Governments, the ERC-COG GemOmics (101044142), and the DFG (DFG LO 2314/1-1).

## Author contributions

**Sruthi Sunil**: Conceptualization; Data curation; Formal analysis; Validation; Investigation; Visualization; Methodology; Writing—original draft. **Simon Beeh**: Conceptualization; Data curation; Formal analysis; Investigation; Visualization; Methodology; Writing—review and editing. **Eva Stöbbe**: Formal analysis; Investigation; Methodology. **Kathrin Fischer**: Data curation; Formal analysis; Investigation; Methodology. **Franziska Wilhelm**: Data curation; Formal analysis; Investigation; Methodology. **Aron Meral**: Formal analysis; Investigation; Methodology. **Celia Paris**: Formal analysis; Investigation; Methodology. **Luisa Teasdale**: Data curation; Formal analysis; Validation; Investigation; Visualization; Methodology. **Zhihao Jiang**: Formal analysis; Validation; Investigation; Visualization; Methodology. **Lisha Zhang**: Formal analysis; Validation; Investigation; Visualization; Methodology. **Moritz Urban**: Data curation; Visualization; Methodology. **Emmanuel Aguilar Parras**: Formal analysis; Investigation. **Thorsten Nürnberger**: Supervision; Funding acquisition; Writing—review and editing. **Detlef Weigel**: Supervision; Funding acquisition; Writing—review and editing. **Rosa Lozano-Duran**: Supervision; Funding acquisition; Writing—review and editing. **Farid El Kasmi**: Conceptualization; Formal analysis; Supervision; Funding acquisition; Investigation; Visualization; Methodology; Writing—original draft; Project administration; Writing—review and editing.

Source data underlying figure panels in this paper may have individual authorship assigned. Where available, figure panel/source data authorship is listed in the following database record: biostudies:S-SCDT-10_1038-S44319-024-00240-4.

## Disclosure and competing interests statement

Detlef Weigel holds equity in Computomics, which advises plant breeders. Detlef Weigel also consults for KWS SE, a globally active plant breeder and seed producer.

# Expanded View Figures

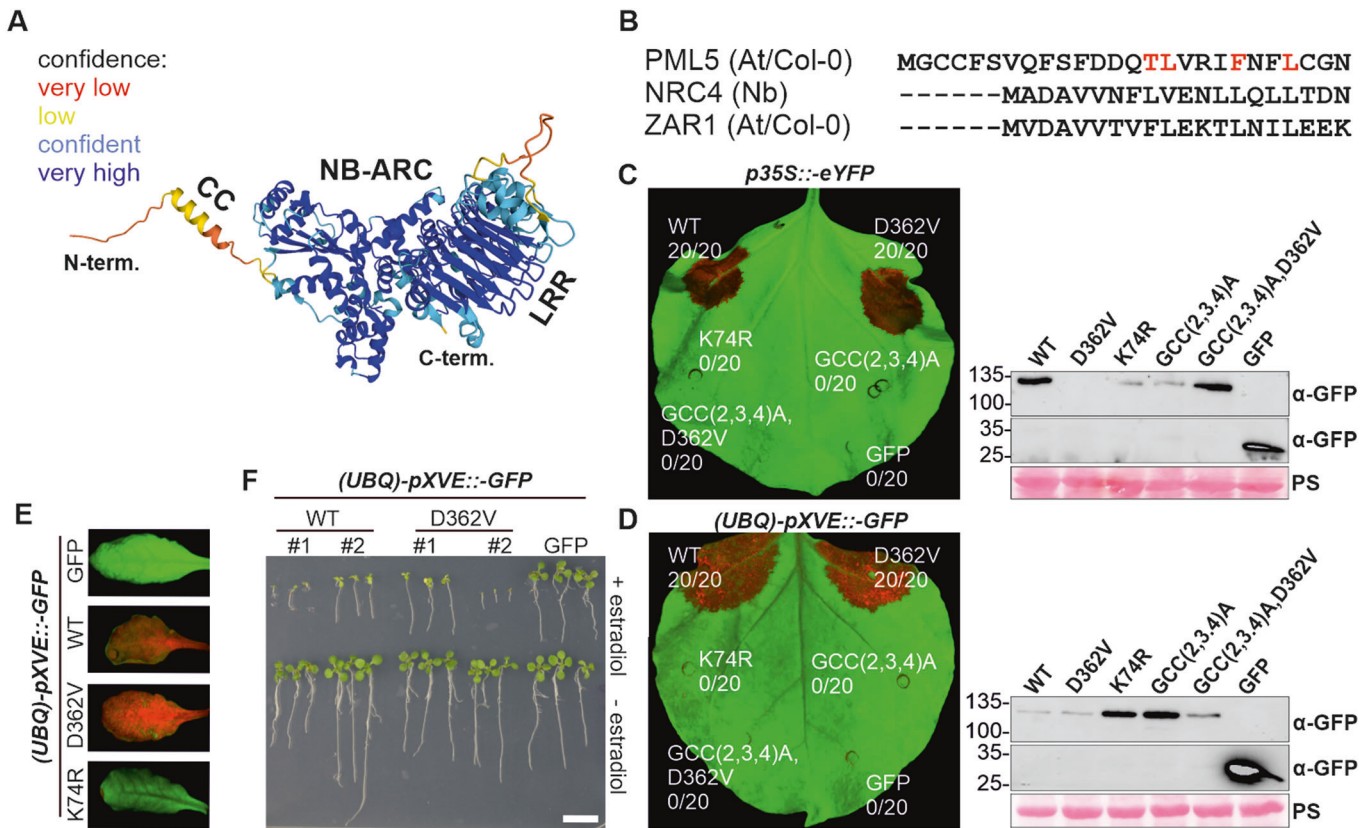

**Figure EV1. PML5 functions as a canonical cell death inducing NLR.**

(A) AlphaFold 2 structural prediction of PML5. CC, NB-ARC, and LRR domains are indicated. A potential alpha 1 helix is predicted between Asp/D13 and Ile/I 27. (B) Sequence alignment of the first 26 N-terminal amino acids of Col-0 PML5, ZAR1 and *N. benthamiana* (Nb) NRC4, highlighting the hydrophobic and polar uncharged amino acids mutated in PML5 (see Fig. 1D). (C, D) Cell death induced by transiently expressed wild-type PML5 and mutant variants (C) constitutively under 35S promoter (top left) or conditionally expressed under a 35S promoter-controlled estradiol inducible system (bottom left). GFP (C) and Citrine-HA (D) served as negative controls. Total protein extracts were immunoblotted and detected with an anti-GFP (α-GFP) antibody. Protein expression analyses of constructs infiltrated are shown on right side with single GFP (C) and Citrine-HA (D) in the respective lower blot around 30 kDa. Cell death images were taken 2 days post-infiltration (C) or 3 days post-induction (D). Leaves are shown in false color, red indicates cell death and green healthy/alive tissue. WT = wild-type PML5; D362V = MHD mutant; K74R = P-loop mutant; GCC(2,3,4) A = N-myristoylation and S-acylation (PTM) mutant; GCC/2,3,4)A,D362V = PTM/MHD quadruple mutant. (E) Cell death phenotype in a single rosette leaf of 4-week-old Arabidopsis plant 18 h post-induction with 20 µM estradiol is shown. Leaves are shown in false color, red indicates cell death in wild-type PML5 and D362V lines, and green healthy/alive tissue in PML5 K74R and GFP negative control lines. (F) Growth restriction/cell death phenotype of two independent transgenic *Arabidopsis* lines conditionally overexpressing PML5-GFP (WT), D362V-GFP. A *35::GFP* plant line was used as a control. 10-day-old Arabidopsis seedlings grown on ½ MS plates supplemented with or without 20 µM estradiol are shown. Scale bars, 1 cm.

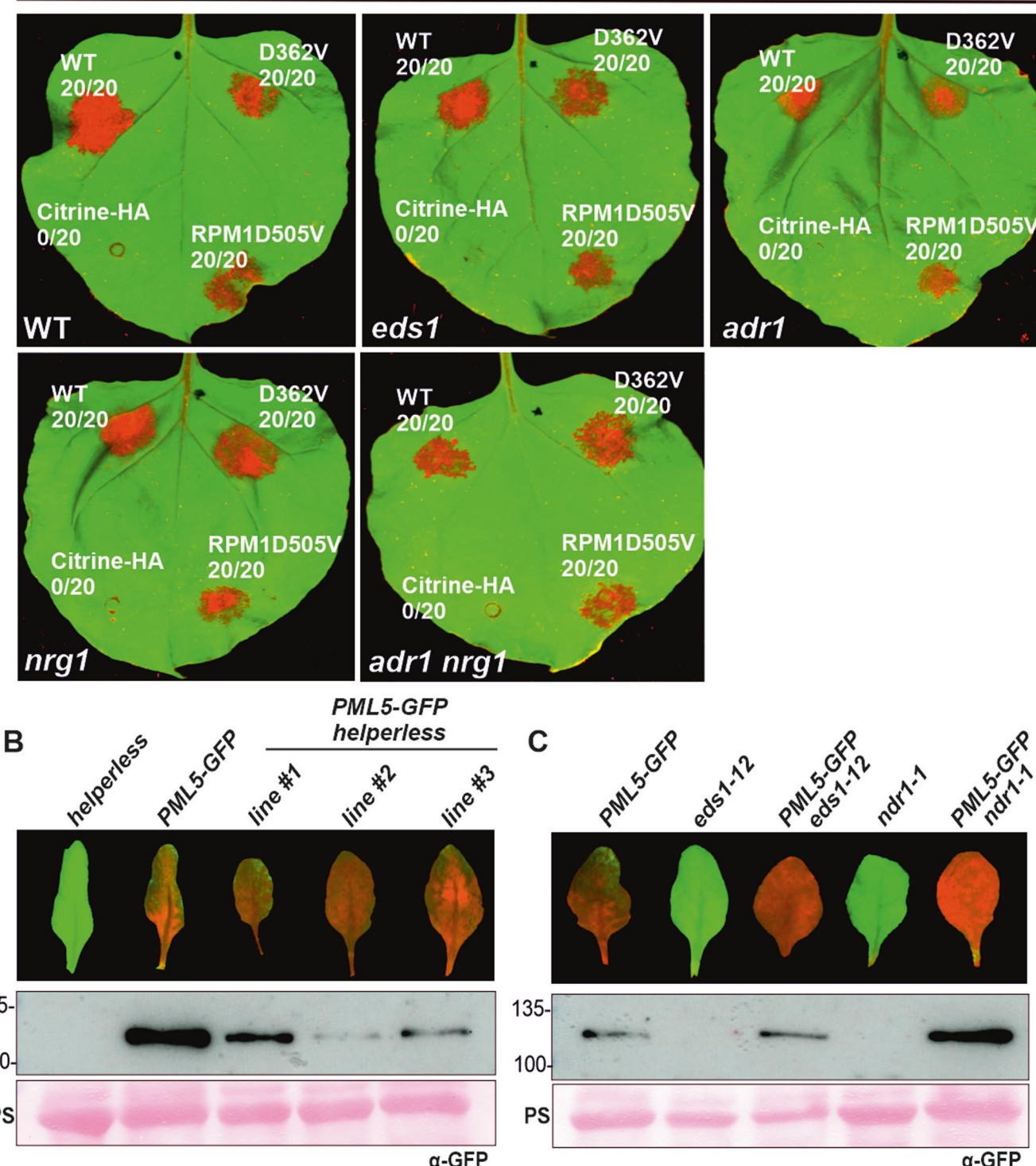

**Figure EV2. PML5-induced cell death is independent of EDS1, RNL helper NLRs, and NDR1.**

(A) Cell death induced by transiently expressed wild-type PML5 (WT) and D362V mutant in WT, *eds1, adr1, nrg1, adr1 nrg1 N. benthamiana* mutant lines. The RPM1 D505V auto-active mutant was used as a positive control and Citrine-HA as a negative control. Images were taken 1 day post-induction with 20 μM estradiol. Leaves are shown in false color, red indicates cell death and green healthy/alive tissue. (B) Cell death in three independent transgenic *Arabidopsis helperless* mutant lines conditionally expressing PML5-GFP. PML5-GFP in Col-0 (PML5-GFP) served as the positive control. Protein blot (bottom) shows the expression of PML5-GFP detected with an anti-GFP (α-GFP) antibody in the positive control and three independent transgenic PML5-GFP *helperless* lines. Ponceau S (PS) staining is shown as a loading control. Images and samples for protein blot were taken 48 h post-induction with 20 μM estradiol. Leaves are shown in false color, red indicates cell death and green healthy/alive tissue. (C) Cell death in transgenic *Arabidopsis eds1-12* and *ndr1-1* mutant conditionally expressing PML5-GFP. PML5-GFP in Col-0 (PML5-GFP) served as the positive control. Protein blot (below) shows the expression of PML5-GFP detected with anti-GFP (α-GFP) antibody in the positive control and transgenic PML5-GFP *eds1-12* and *ndr1-1* mutants. Ponceau S (PS) staining is shown as a loading control. Samples for protein blot were taken 18 h post-induction (hpi) with 20 μM estradiol, cell death images were taken 48 hpi. Leaves are shown in false color, red indicates cell death and green healthy/alive tissue.

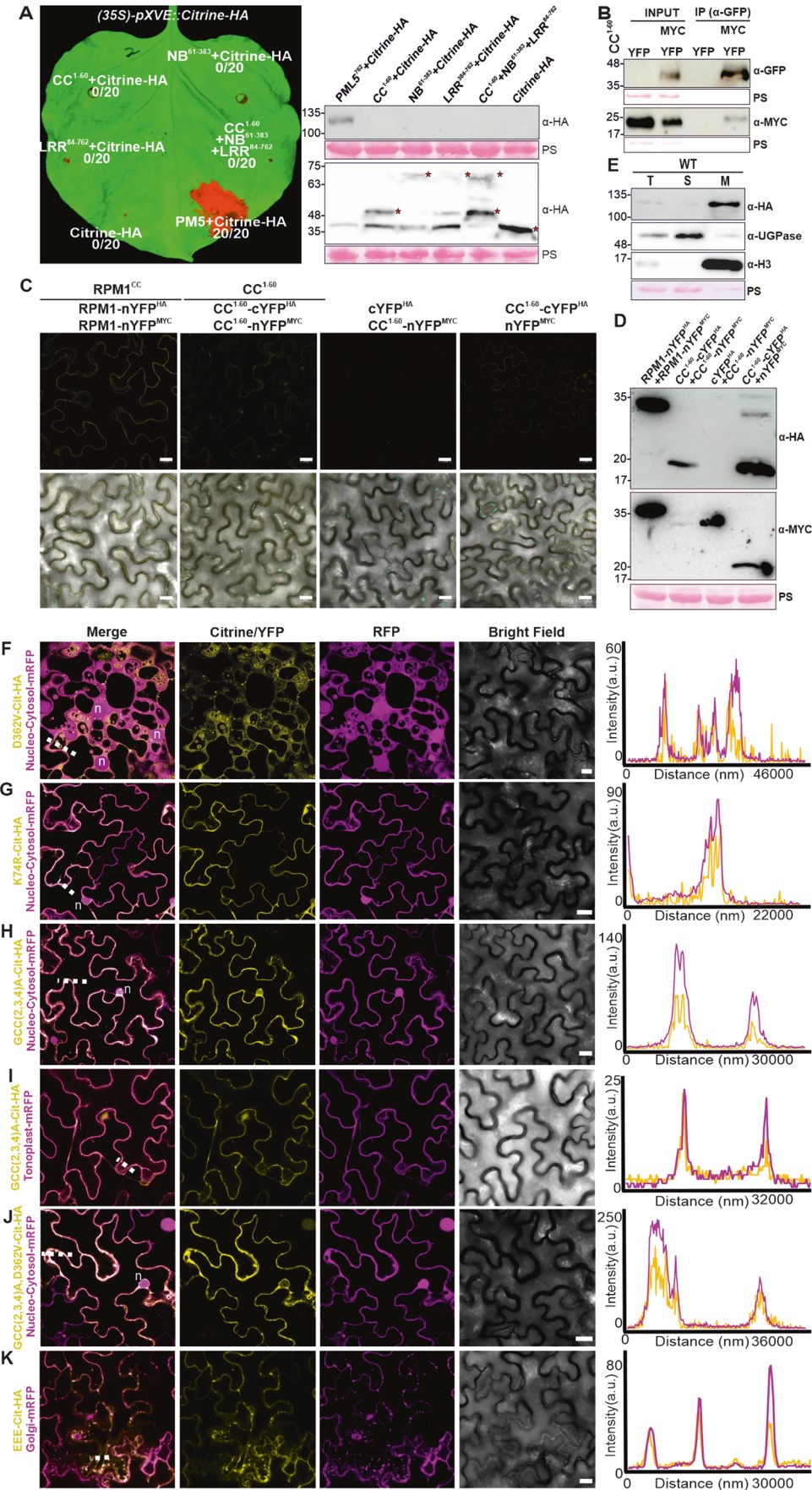

◀ **Figure EV3.  PML5 N-terminal 60 amino acids are sufficient for cell death induction.**

(A) Combination of PML5 CC$^{1-60}$, NB$^{61-383}$, and LRR$^{384-762}$ domains is not sufficient to induce a WT-like cell death response. PML5 domains were transiently expressed alone or in co-infiltrations in *N. benthamiana*. Image was taken 2 days post-induction with 20 µM estradiol. (Right panel) Total protein extracts were immunoblotted and detected with an anti-HA (α-HA) antibody. Ponceau S (PS) staining is shown as a loading control. The red asterisks indicate the different fragments expressed. Leaves are shown in false color, red indicates cell death and green healthy/alive tissue. (B) PML5 CC$^{1-60}$ self-associates in *N. benthamiana* transient expression. C-terminally YFP and MYC tagged CC$^{1-60}$ were co-expressed. Total proteins were immunoprecipitated using anti-GFP (α-GFP) beads and immunoblotted using an anti-GFP (α-GFP) and anti-Myc (α-myc) antibody. Ponceau S (PS) staining is shown as a loading control. (C) Bimolecular fluorescence complementation (BiFC) experiment of PML5 CC$^{1-60}$ fusion proteins, CC$^{1-60}$-cYFP$^{HA}$ and CC$^{1-60}$-nYFP$^{MYC}$, transiently expressed in *N. benthamiana*. No YFP complementation different to the negative control could be detected for the PML5 CC$^{1-60}$ domain. RPM1 CC$^{1-155}$ served as positive control showing YFP complementation. Co-expression of cYFP$^{HA}$ and nYFP$^{MYC}$ with PML5 CC$^{1-60}$-nYFP$^{MYC}$ and PML5 CC$^{1-60}$-cYFP$^{HA}$, respectively, served as negative controls. Scale bars, 20 µm. (D) Protein-blot analysis of total proteins from the transiently expressed proteins of the BiFC assay shown EV3C. Proteins were detected using anti-Myc (α-MYC) and anti-HA (α-HA) antibodies. Ponceau S (PS) staining is shown as a loading control. (E) Subcellular fractionation of transiently expressed PML5-Citrine-HA protein indicates a strong microsomal/membrane association. Total protein extracts were immunoblotted and detected with an anti-HA (α-HA) antibody for PML5, anti-UGPase (α-UGPase) as a cytosolic marker, and anti-Histone H3 (α-Histone H3) as a microsomal marker. T = total protein fraction; S = soluble protein fraction; M = microsomal protein fraction. (F–K) Confocal laser scanning microscopy showing subcellular localization of transiently expressed PML5-Citrine-HA mutants in comparison to different marker proteins: (F) D362V with nucleo-cytoplasmic mRFP; (G) K74R with nucleo-cytoplasmic mRFP; (H, I) GCC(2,3,4)A with nucleo-cytoplasmic mRFP and tonoplast mRFP; (J) GCC(2,3,4)A,D362V with nucleo-cytoplasmic mRFP; (K) T15E,L16E,F20E (EEE) with Golgi localized mRFP. White dotted lines indicate the area used for colocalization profile analysis and the corresponding profiles are shown on the right. *n* = nucleus. Scale bars, 10 µm; for information on compartment marker proteins, see the Methods section. Protein expression was induced with 20 µM estradiol and confocal imaging was performed at 6–8 h post-induction.

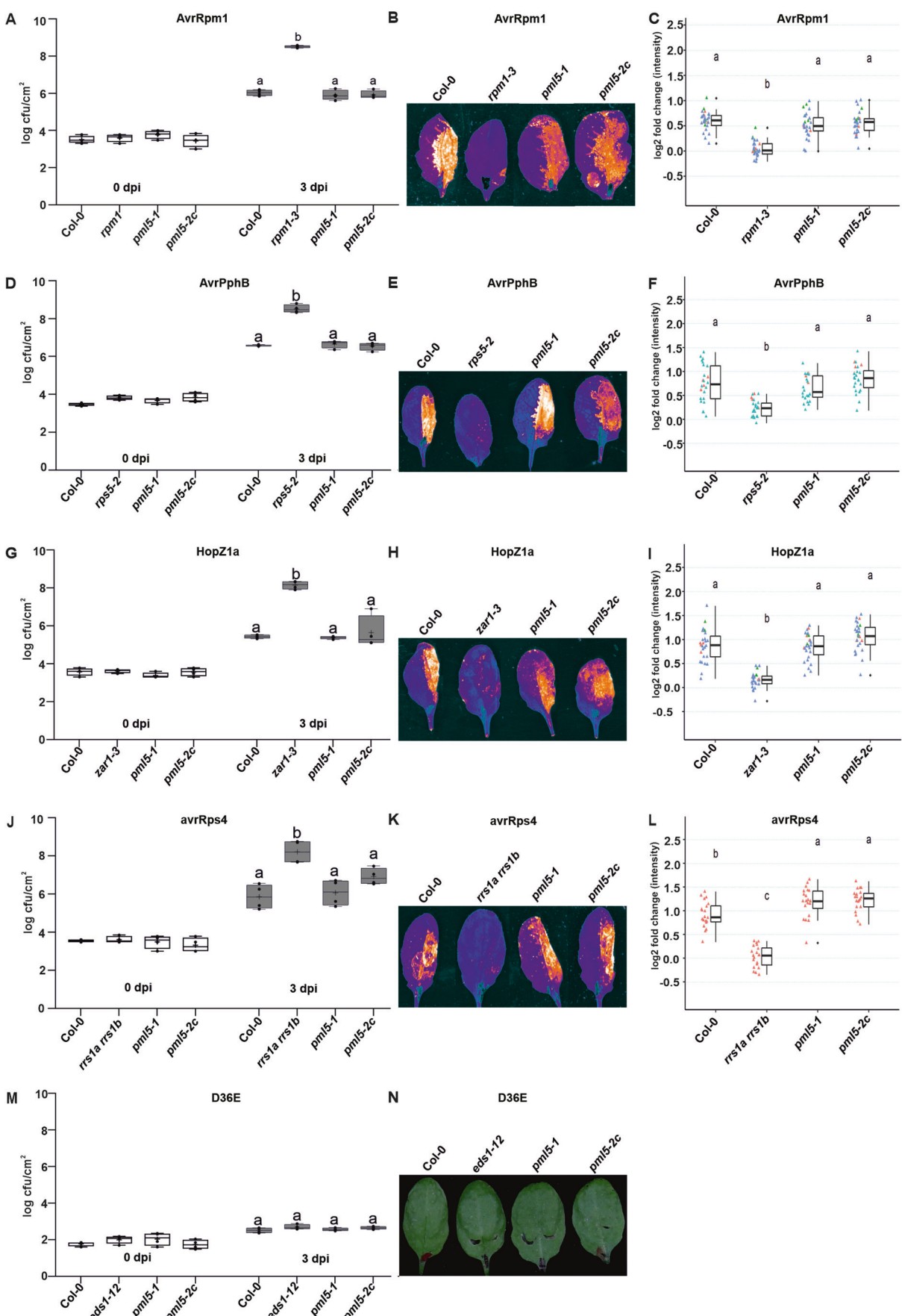

**Figure EV4. *PML5* does not contribute to resistance against avirulent *Pst* DC3000.**

(A–M) Resistance and cell death induced upon infection with different avirulent *Pst* DC3000 strains is not altered in *pml5* mutants. Rosette leaves of 6-week-old *Arabidopsis* plants were hand infiltrated with various *Pst* DC3000 strains using an $OD_{600} = 0.001$: (A) *Pst* DC3000 *AvrRpm1*, (D) *Pst* DC3000 *AvrPphB*, (G) *Pst* DC3000 *HopZ1a*, (J) *Pst* DC3000 *AvrRps4*, (M) *Pst* DC3000 D36E and bacterial growth was determined on day 0 and day 3 post infiltration. For (A, D, G, J, M), data are shown as boxplots and data points (colony forming units per square cm—$cfu/cm^2$) are indicated as black dots and represent 1 biological replicate with 4 technical replicates each ($n = 4$, with 6 leaf samples each). (B, E, H, K) Induction and (C, F, I, L) strength of cell death is not affected in *pml5* mutants. The right side of the leaves was hand infiltrated with *Pst* DC3000: *AvrRpm1* (B, C), *AvrPphB* (E, F), and *HopZ1a* (H, I) at an $OD_{600} = 0.1$ and with *Pf0*-1 *AvrRps4* (K, L) at an $OD_{600} = 0.2$. Infiltrated leaves were imaged with a Typhoon laser scanner 5 h post infiltration (hpi) for *AvrRpm1* (B), 22 hpi for *AvrPphB* (E), 24 hpi for *HopZ1a* (H), and *AvrRps4* (K). The leaf images are shown in false color: Purple/blueish parts indicate non-infiltrated healthy tissue, and orange/yellowish dead cells. (C, F, I, L) Quantification of cell death intensity measured of infiltrated leaves similar to leaves shown in (B, E, H, K). Details on the methodology can be found in Methods section. Data are presented as boxplots. Results shown are from 1 biological replicate with 20 technical replicates for (L), 2 biological replicates with 24 technical replicates for (F) and 3 biological replicates with 28 technical replicates for (C, I). Data points of the different biological replicates are indicated by differently colored triangles. (N) Infiltration of *Pst* DC3000 D36E does not induce visible disease symptoms on *pml5-1*, *pml5-2c* or Col-0 and *eds1-12*. Data information: For (A, D, G, J, M) data are presented as boxplots (center line, median; bounds of box, the first and the third quartiles; whiskers, 1.5 times the interquartile range; error bar, minima and maxima), similarly for (C, F, I, L). There, black dots additionally represent outliers. Data points with different letters indicate significant differences of $P \leq 0.05$ (one-way ANOVA with a post hoc Tukey's HSD test). Exact $P$ values for all experiments are provided in Dataset EV1.

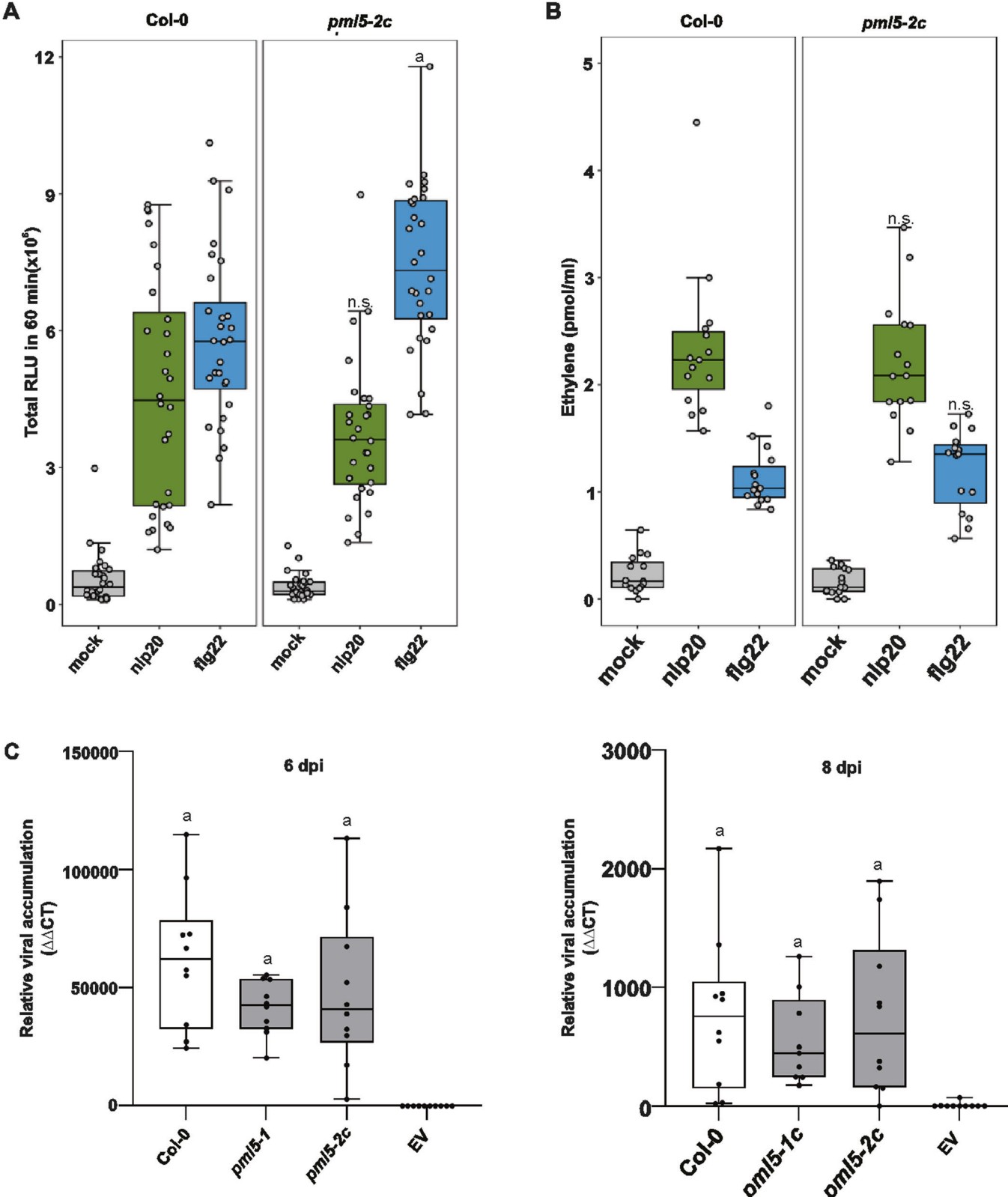

Figure EV5. Loss of *PML5* has no effect on PRR-induced responses and TRV resistance.

(A) Total ROS production in leaf discs of Col-0 and *pml5-2c* mutant treated with water (mock), 1 µM nlp20, or 100 nM flg22 over 60 min. RLU, relative light unit. Data points are indicated as gray dots from four biological replicates each with seven technical replicates ($n = 28$) and plotted as boxplots. The gray color box indicates mock control, green indicates nlp20 treatment and blue indicates flg22 treatment in Col-0 and *pml5-2c* plants. (B) Ethylene accumulation in Col-0 and *pml5* mutant after 4 h treatment with water (mock), 1 µM nlp20, or 1 µM flg22. Data points are indicated as gray dots from five biological replicates each with 3 technical replicates ($n = 15$). (C) TRV viral accumulation was determined by qRT-PCR at 6 days post infection (dpi) (left) and 8 dpi (right) in infected Col-0, *pml5-1c*, and *pml5-2c* plants. Col-0 infected with an Empty Vector (EV) control served as negative control. Data points are from 10 different independent plants and the experiment was repeated 3 times (3 technical replicates) with similar results. Data information: Data in (A–C) are represented as boxplots (center line, median; bounds of box, the first and the third quartiles; whiskers, 1.5 times the interquartile range; error bar, minima and maxima). Statistical differences compared to the same treatment (A, B) in Col-0 were analyzed by two-sided Student's t-test ($a = 0.05$) and are indicated with letters (a, $P < 0.01$, 'n.s.' not statistically different). In (C), data points with different letters indicate significant differences of $P \leq 0.05$ (one-way ANOVA with a post hoc Tukey's HSD test). Exact $P$ values for all experiments are provided in Dataset EV1.

