## [Peer Review File · EMBO Reports]

Activation of an atypical plant NLR with an N-terminal deletion initiates cell death at the vacuole

Sruthi Sunil, Simon Beeh, Eva Stöbbe, Kathrin Fischer, Franziska Wilhelm, Celia Paris, Aron Meral, Luisa Teasdale, Zhihao Jiang, Lisha Zhang, Moritz Urban, Emmanuel Parras, Thorsten Nürnberger, Detlef Weigel, Rosa Lozano-Duran, and Farid El Kasmi

Corresponding author: Farid El Kasmi (farid.el-kasmi@zmbp.uni-tuebingen.de)

Review Timeline:

Submission Date:	27th Oct 23
Editorial Decision:	21st Nov 23
Revision Received:	24th May 24
Editorial Decision:	21st Jun 24
Revision Received:	26th Jul 24
Accepted:	12th Aug 24

Editor: Achim Breiling

Transaction Report:

Dear Dr. El Kasmi,

Thank you for the submission of your research manuscript to EMBO reports. I have now received the reports from the three referees that were asked to evaluate your study, which can be found at the end of this email.

As you will see, the referees think that the findings are of interest. However, they have several comments, concerns, and suggestions, indicating that a major revision of the manuscript is necessary to allow publication of the study in EMBO reports. As the reports are below, and all the referee concerns need to be addressed, I will not detail them here.

Given the constructive referee comments, I would like to invite you to revise your manuscript with the understanding that all referee concerns must be addressed in the revised manuscript or in a detailed point-by-point response. Acceptance of your manuscript will depend on a positive outcome of a second round of review. It is EMBO reports policy to allow a single round of revision only and acceptance of the manuscript will therefore depend on the completeness of your responses included in the next, final version of the manuscript.

- 1) a .docx formatted version of the final manuscript text (including legends for main figures, EV figures and tables), but without the figures included. Figure legends should be compiled at the end of the manuscript text.
- 2) individual production quality figure files as .eps, .tif, .jpg (one file per figure), of main figures (up to 8) and EV figures. Please upload these as separate, individual files upon re-submission.

- 4) a complete author checklist, which you can download from our author guidelines (<https://www.embopress.org/page/journal/14693178/authorguide>). Please insert page numbers in the checklist to indicate where the requested information can be found in the manuscript. The completed author checklist will also be part of the RPF.

- 5) that primary datasets produced in this study (e.g. RNA-seq, ChIP-seq, structural and array data) are deposited in an

appropriate public database. If no primary datasets have been deposited, please also state this in a dedicated section (e.g. 'No primary datasets have been generated and deposited'), see below.

The accession numbers and database should be listed in a formal "Data Availability" section (placed after Materials & Methods) that follows the model below. This is now mandatory (like the COI statement). Please note that the Data Availability Section is restricted to new primary data that are part of this study. This section is mandatory. As indicated above, if no primary datasets have been deposited, please state this in this section

Data availability

8) Regarding data quantification and statistics, please make sure that the number "n" for how many independent experiments were performed, their nature (biological versus technical replicates), the bars and error bars (e.g. SEM, SD) and the test used to calculate p-values is indicated in the respective figure legends (also for potential EV figures and all those in the final Appendix). Please also check that all the p-values are explained in the legend, and that these fit to those shown in the figure. Please provide statistical testing where applicable. Please avoid the phrase 'independent experiment', but clearly state if these were biological or technical replicates. Please also indicate (e.g. with n.s.) if testing was performed, but the differences are not significant. In case n=2, please show the data as separate datapoints without error bars and statistics. See also: <http://www.embopress.org/page/journal/14693178/authorguide#statisticalanalysis>

9) Please add scale bars of similar style and thickness to all the microscopic images, using clearly visible black or white bars (depending on the background). Please place these in the lower right corner of the images themselves. Please do not write on or near the bars in the image but define the size in the respective figure legend.

10) Please also note our reference format:

12) We now use CRedit to specify the contributions of each author in the journal submission system. CRedit replaces the author contribution section. Please use the free text box to provide more detailed descriptions and do not provide your final manuscript text file with an author contributions section. See also our guide to authors: <https://www.embopress.org/page/journal/14693178/authorguide#authorshipguidelines>

13) We would encourage you to use 'Structured Methods', our new Materials and Methods format. According to this format, the

Materials and Methods section should include a Reagents and Tools Table (listing key reagents, experimental models, software and relevant equipment and including their sources and relevant identifiers) followed by a Methods and Protocols section in which we encourage the authors to describe their methods using a step-by-step protocol format with bullet points, to facilitate the adoption of the methodologies across labs. More information on how to adhere to this format as well as downloadable templates (.doc or .xls) for the Reagents and Tools Table can be found in our author guidelines (section 'Structured Methods'):

14) Please order the manuscript sections like this, using these names:

Title page - Abstract - Keywords - Introduction - Results - Discussion - Materials and Methods - Data availability section - Acknowledgements - Disclosure and Competing Interests Statement - References - Figure legends - Expanded View Figure legends

I look forward to seeing a revised version of your manuscript when it is ready. Please let me know if you have questions or comments regarding the revision.

Please use this link to submit your revision: <https://embor.msubmit.net/cgi-bin/main.plex>

Yours sincerely,

Referee #1:

Sunil et al. reported an atypical ancient and autonomous CC[G10/GA]-type NLR (ANL) immune receptor named PML5, which is characterized by a conserved deletion in its N-terminal CC signaling domain. The authors tried to unravel the structure-function relationship of PML5, with an emphasis on its capacity to trigger cell death despite a 113 amino acid deletion. To dissect the function of PML5, the authors applied a range of techniques including mutagenesis, co-immunoprecipitation, Blue Native PAGE, and confocal microscopy. Their findings demonstrate that despite its structural uniqueness, PML5 retains the cell-death initiation functionality typical of a full-length ANL. This research advances our understanding of the diversity and evolutionary adaptability of plant immune receptors. The methods they used in their experiments provide good support for their findings. Nonetheless, certain areas of the study could benefit from further experimental exploration or revisions in the text to enhance the clarity and the overall comprehensiveness of their findings.

My major concerns:

1. Please consider validating the hypothesized mechanism of PML5-induced cell death, particularly focusing on its potential role in vacuolar fragmentation and calcium influx. This could involve more detailed biochemical or biophysical studies.
2. It would be beneficial to further explore the evolutionary aspect of PML5. Consider conducting a comparative analysis with more distant species or a broader range of Brassicales to understand the evolutionary trajectory and significance of the observed deletions in PML5.
3. In the Abstract, please consider providing a clearer context of the study's significance within the broader field of plant immunity and explicitly stating the hypothesis or research question.
4. The Discussion section could benefit from a more elaborate comparison with existing literature, particularly in the context of PML5's unique features compared to other CNLs and ANLs.

Some minor points for improvement:

Lines 25-41: Please consider revising the abstract to more explicitly highlight the novelty and implications of your findings.

Lines 154-251: Please clarify the experimental methods used, especially for mutagenesis and imaging studies, and ensure that the significance of each finding is clearly articulated.

Lines 415-521: Please expand on the implications of your findings in the broader context of plant immunity and NLR function. Additionally, a more detailed discussion on the potential evolutionary significance of PML5 would enhance the depth of the manuscript.

Throughout the manuscript, please ensure consistency in terminology and acronym usage for clarity:

1. If the term "nucleotide-binding leucine-rich repeat receptors" is introduced as "NLRs," consistently use "NLRs" throughout the manuscript after the first instance. Avoid switching to alternative acronyms like "NB-LRRs" unless specifically discussing a different context or classification.
2. When first mentioning "Potentially Membrane Localized NLRs," it is abbreviated as "PML NLRs." Stick with "PML NLRs"

consistently in the manuscript rather than introducing new abbreviations or reverting to the full name.

3. The term "co-immunoprecipitation" is often abbreviated as "Co-IP." Once defined, consistently use "Co-IP" throughout your manuscript to refer to this technique.

4. For the protein "PML5," if you refer to mutants or variations, such as "PML5-Citrine-HA," "PML5 D362V," or "PML5 K74R," maintain this naming convention. Do not switch to variations like "Citrine-HA-tagged PML5" or "PML5-D362V" unless there's a specific reason that needs to be highlighted.

5. If you introduce "Blue Native PAGE" as "BN-PAGE," continue using "BN-PAGE" in all subsequent mentions. Avoid interchanging it with "Blue Native Polyacrylamide Gel Electrophoresis" to prevent confusion.

6. When discussing post-translational modifications, if you introduce "N-myristoylation" and "S-acylation" and abbreviate them later in the text, ensure that the abbreviations are clear and used consistently.

Please consider a thorough review for grammatical and syntactical improvements to enhance readability:

Lines 25-41: Please consider revising the sentence structure for clarity. For example, "Activated coiled-coil (CC) domain containing NLRs (CNLs) oligomerize to form apparent cation channels promoting calcium influx and cell death, with the alpha-1 helix of the individual CC domains penetrating membranes." could be rephrased for clarity to: "CNLs with activated CC domains oligomerize and form channels that appear to facilitate calcium influx and cell death, with the alpha-1 helix of each CC domain penetrating the membrane."

Lines 154-251: The passage "We identified 20 CNLs in Arabidopsis Col-0 with putative N-myristoylation and/or S-acylation sites at their very N-terminus (Table EV1 and Fig 1A), which we named Potentially Membrane Localized (PML) NLRs." might be clearer as: "In Arabidopsis Col-0, we identified 20 CNLs with putative N-myristoylation and S-acylation sites at their N-termini, which we have termed Potentially Membrane Localized (PML) NLRs (Table EV1 and Fig 1A)."

Lines 370-373: The sentence "As reported above, induced (over-)expression of active/activated PML5 strongly induced cell death and altered vacuolar morphology (Fig 1 and 4), suggests that PML5 functions as a canonical CNL but with some specific characteristics - a dynamic Golgi/tonoplast localization and the vacuolar fragmentation phenotype." could be improved for readability by adjusting it to: "As previously mentioned, the induced overexpression of active PML5 markedly induces cell death and alters vacuolar morphology (Fig 1 and 4), indicating that PML5, while functioning as a canonical CNL, also exhibits distinct characteristics, such as dynamic localization to Golgi and tonoplast, and a pronounced vacuolar fragmentation phenotype."

Figure 5: Extended figure 9: green text and green y and x axis look awkward and light green hard to read on white screen.

Figure 1 Legend: Units like 80µM sometimes written together instead of 80 µM (SI Units style convention)

Extended Figure 1: Phylo tree text hard to read (Use pdf instead of png)

Figure 5 Legend: Line 906 has an alpha value but Line 895 does not have

Line 934: "upper panle" -> panel

Line 968: NBARC => NB-ARC

Extended Figure 10 A: Use of 3 colors but not mentioning what their meanings are?

Line 1047: Error in legend: (A) Pst DC3000 is wrong according to the image it's Pst DC3000 avrRPM1.

Referee #2:

Sunil and colleagues present a detailed dissection of the atypical plant immune receptor PML5 in Arabidopsis. They nicely demonstrate the autonomous activity of the CNL/ANL-type receptor and importantly demonstrate novel localization to the tonoplast/vacuole and conserved biochemical functions in oligomerization and N-terminal motif-mediated pore formation and cell death. This is a complete study with a solid and appropriate breadth of experimentation that addresses the majority of questions surrounding the evolution and function of a new/atypical immune receptor architecture. I am highly supportive of this work overall and have only a few minor comments/suggestions.

Main Comments:

- Given that wildtype PML5 causes cell death on its own, it seems relevant to explore its transcriptional regulation during stress. Is PML5 generally upregulated during basal immune responses, ETI, or other stresses? Any correlation to stresses leading to vacuolar fragmentation/cell death?

- Fig EV9: The false-colored images appear to show decreased HR intensity in pml5 mutants (more so in pml5-2c) for CNL but not TNL-mediated responses. Did the authors explore this? Quantification of several images (pixel intensity), trypan blue staining, or ion leakage assays could be helpful here. Do you think that PML5 potentiates cell death activation to give a more robust HR? If so, are PML5 transcripts induced during interactions with virulent or avirulent Pst?

Minor Comments:

-Lines 330-341: Really nice data. Do these sizes make biological sense for distinct sizes of oligomers? Are you seeing pentamers or even higher complexes? Is it possible or informative to model the oligomer using AF2?

-Fig 4B: Tonoplast marker is hard to interpret, looks like signal everywhere. Is there a better marker for this?

-Fig EV8 and localization data: Is there an issue localizing PML5-mCitirne-HA at the membrane in confocal experiments vs fractionation? Why do you think this is happening and is the tag interfering with the process? One suggestion is to perform localization experiments using the N-term motif mutants that lose cell death (L-to-E in N-terminal motif) but are presumably still oligomerizing as is described and exploited by NRCs. This might improve imaging.

-Line 39: typo "PML5-ike"

-Fig 5D: The lines flowered early, but it would be useful to mention the timepoint that this happened at, and the relative difference to WT. Did they flower much earlier, or just a week earlier?

-Line 772: KB 'agarose' plates. Agarose is mentioned a few times and should be corrected to agar.

-grammar in the title. 'An atypical...'

Referee #3:

This manuscript by Sunil et al provides a large amount of data that collectively support that *Arabidopsis thaliana* PML5, an atypical NLR with a deletion in in the CC domain, functions as a canonical NLR. Comparative genomics revealed that such a deletion in CC domain is not restricted to *A. thaliana* but is present in other Brassicales plants. The authors then showed that PML5 has ability to oligomerize, to increase cytosolic Ca concentration, and to induce cell death. Interestingly, the authors found the localization to Golgi and tonoplast is a unique property of PML5, because other ANL class of NLRs have been reported to localize plasma membrane via post-translational modifications such as N-myristoylation and S-acylation. Pathoassays using a T-DNA insertional mutant and a newly-created CRISPR mutant support a role of PML5 in basal immunity against *Pseudomonas syringae*.

Overall, the conclusion of this manuscript is supported well by the solid data, and the writing is clear and easy to follow. The novelty resides in the discovery of the unique mode of cell death induction by PML5. I have only a few comments that may further improve this nice manuscript.

Major points:

1. The authors argue that PML5 induces vacuolar fragmentation that correlate with cell death induction. I think this claim needs a different line of evidence or through examination. In Fig. 4B, the localization pattern of the vacuole marker protein is unaltered. How can this be explained? Is vacuolar fragmentation specific to PML5-induced cell death? How can you rule out the possibility that the vacuolar fragmentation is an artefact due to PML5 overexpression? Have you tested whether PML5 driven by native promoter induces vacuolar fragmentation? It could also be that vacuolar fragmentation is a consequence of cell death not an active regulation by PML5.

2. Are the GCC residues required for PML5 localization to Golgi membrane and the tonoplast? No data are provided testing colocalization of GCC(2, 3,4)A-Ct-HA and the Golgi or Vacuole marker proteins.

Other comments:

1. Is PML5 transcriptionally induced upon pathogen challenge? If this is the case, it would support the role of PML5 in immunity, as transcriptional induction of PML5 is sufficient to induce cell death (Fig. 1E and Fig EV3E). Along the same line, have you tested whether pre- or co-treatment of estradiol with *P. syringae* enhances bacteria resistance in the transgenic lines used in Fig EV4E?

2. Fig EV6: RPMI is labeled 0/20, although I see clear cell death in the infiltrated area.

3. Fig EV10: Statistical analysis should be applied.

EMBOR-2023-58376V1**Reviewers' reports and detailed point-by-point responses by authors**

We thank all three reviewers for their recognition of our work's significance. We have carefully considered their constructive suggestions and have made significant revisions to the manuscript. These revisions include additional experimental results, responses to all reviewer questions, and updated figures and text. We have attached a change-tracked version and a clean version of the manuscript for the reviewers' convenience. We hope the revised manuscript satisfies the reviewers and they will agree that it is suitable for publication in the EMBO Reports.

Referee #1:

Sunil et al. reported an atypical ancient and autonomous CC[G10/GA]-type NLR (ANL) immune receptor named PML5, which is characterized by a conserved deletion in its N-terminal CC signaling domain. The authors tried to unravel the structure-function relationship of PML5, with an emphasis on its capacity to trigger cell death despite a 113 amino acid deletion. To dissect the function of PML5, the authors applied a range of techniques including mutagenesis, co-immunoprecipitation, Blue Native PAGE, and confocal microscopy. Their findings demonstrate that despite its structural uniqueness, PML5 retains the cell-death initiation functionality typical of a full-length ANL. This research advances our understanding of the diversity and evolutionary adaptability of plant immune receptors. The methods they used in their experiments provide good support for their findings. Nonetheless, certain areas of the study could benefit from further experimental exploration or revisions in the text to enhance the clarity and the overall comprehensiveness of their findings.

We appreciate the reviewer's concise perception of our study and the appreciation for it.

My major concerns:

1. Please consider validating the hypothesized mechanism of PML5-induced cell death, particularly focusing on its potential role in vacuolar fragmentation and calcium influx. This could involve more detailed biochemical or biophysical studies.

We thank the reviewer for these suggestions. We now have included further experiments that support our conclusion that PML5 induces cell death by potentially forming calcium permeable channels (resistosomes) at the tonoplast. We not only show induced calcium influx by expression of PML5 D362V (Fig 1F) but also show that the general calcium channel inhibitor lanthanum (LaCl₃) can block PML5 induced cell death (new Fig 1G). These experiments strongly suggest that active PML5 functions as a canonical CNL forming resistosomes leading to calcium fluxes. Further, do we now present confocal imaging data clearly showing co-localization of cell-death active PML5 with our newly generated tonoplast marker (Fig 4C) and co-localization of wild-type PML5 in stable transgenic Arabidopsis lines with tonoplast localized FM4-64 (new Fig 4G). A potential specific role or function in vacuolar fragmentation by PML5 is also supported by the observation that other cell death inducing NLRs or domains of NLRs do not cause a similar phenotype in transient expression assays as addressed in response to major comment of reviewer #3 (see below reviewer figure 3). We do hope that we sufficiently addressed this concern and do want to emphasize that in this study we wanted to focus on a first characterization of this atypical NLR with its unexpected canonical cell death function. A detailed analysis of the specific molecular mechanism leading to this interesting phenotype is part of our current and future investigations and we hope to be able to publish sometime in the near future.

2. It would be beneficial to further explore the evolutionary aspect of PML5. Consider conducting a comparative analysis with more distant species or a broader range of Brassicales to understand the

evolutionary trajectory and significance of the observed deletions in PML5.

This is a valid point raised by the reviewer. In the scope of this study the focus was laid on the first characterization of this very interesting and unusual truncated NLR and to elucidate the potential PML5 cell death function. We did a thorough examination of all plant proteomes available to us and only found a similar deletion in NLRs (ANLs) in Brassicales species presented in this manuscript. We do agree that in the future it would be interesting to get a better understanding of the evolutionary trajectory and significance of the observed deletion.

3. In the Abstract, please consider providing a clearer context of the study's significance within the broader field of plant immunity and explicitly stating the hypothesis or research question.

We thank the reviewer for this comment. Given the limited number of words allowed in the abstract we tried to condense the context and the broader significance as good as possible. However, we now have changed the last sentence of the abstract to better (at least in our opinion) convey the major finding interesting for the broader field of plant immunity. Further, the two other reviewers had no issue with the language, grammatical and syntactical representation of the text, and thus, we anticipate that the text is now, with the suggestions made by this reviewer, very much improved.

4. The Discussion section could benefit from a more elaborate comparison with existing literature, particularly in the context of PML5's unique features compared to other CNLs and ANLs.

We thank the reviewer for this suggestion. We now have thoroughly revised the text including the discussion. We have considered recently published literature in our interpretation of the results and hope to better convey the significance of our findings. We would also like to mention that the two other reviewers very much appreciated our writing and interpretation of the data.

Some minor points for improvement:

Lines 25-41: Please consider revising the abstract to more explicitly highlight the novelty and implications of your findings.

See response to major comment #3.

Lines 154-251: Please clarify the experimental methods used, especially for mutagenesis and imaging studies, and ensure that the significance of each finding is clearly articulated.

We appreciate this comment/suggestion made. We have revised the specific text to better clarify the approaches used and have tried to articulate more precisely the significance of our findings at this paragraph and in general throughout the results section.

Lines 415-521: Please expand on the implications of your findings in the broader context of plant immunity and NLR function. Additionally, a more detailed discussion on the potential evolutionary significance of PML5 would enhance the depth of the manuscript.

We thank the reviewer for this comment and would like to point to the response given to the same reviewer's comment # 4.

Throughout the manuscript, please ensure consistency in terminology and acronym usage for clarity:
1. If the term "nucleotide-binding leucine-rich repeat receptors" is introduced as "NLRs," consistently use "NLRs" throughout the manuscript after the first instance. Avoid switching to alternative acronyms like "NB-LRRs" unless specifically discussing a different context or classification.

Thanks for bringing this up. We have carefully looked through our revised text and were not able to find any switching between NLR and NB-LRR, unless where we specifically refer to the individual domains.

2. When first mentioning "Potentially Membrane Localized NLRs," it is abbreviated as "PML NLRs." Stick with "PML NLRs" consistently in the manuscript rather than introducing new abbreviations or reverting to the full name.

We appreciate this comment. Throughout the revised text we have made sure that we only use PMLs for "Potentially Membrane Localized NLRs" after the abbreviation had been introduced.

3. The term "co-immunoprecipitation" is often abbreviated as "Co-IP." Once defined, consistently use "Co-IP" throughout your manuscript to refer to this technique.

Thank you. We now consistently use Co-IP to refer to this technique throughout our revised text.

4. For the protein "PML5," if you refer to mutants or variations, such as "PML5-Citrine-HA," "PML5 D362V," or "PML5 K74R," maintain this naming convention. Do not switch to variations like "Citrine-HA-tagged PML5" or "PML5-D362V" unless there's a specific reason that needs to be highlighted.

We thank the reviewer for bringing this issue up. We now have used a consistent naming throughout the text and only switched where required.

5. If you introduce "Blue Native PAGE" as "BN-PAGE," continue using "BN-PAGE" in all subsequent mentions. Avoid interchanging it with "Blue Native Polyacrylamide Gel Electrophoresis" to prevent confusion.

We took care of making sure to introduce the abbreviation BN-PAGE and sticking to using only BN-PAGE only once in the following text.

6. When discussing post-translational modifications, if you introduce "N-myristoylation" and "S-acylation" and abbreviate them later in the text, ensure that the abbreviations are clear and used consistently.

We thank the reviewer for bringing this up. In the revised text we only use the terms S-acylation and N-myristoylation and only refer to post translational modifications in general.

Please consider a thorough review for grammatical and syntactical improvements to enhance readability:

Lines 25-41: Please consider revising the sentence structure for clarity. For example, "Activated coiled-coil (CC) domain containing NLRs (CNLs) oligomerize to form apparent cation channels promoting calcium influx and cell death, with the alpha-1 helix of the individual CC domains penetrating membranes." could be rephrased for clarity to: "CNLs with activated CC domains oligomerize and form channels that appear to facilitate calcium influx and cell death, with the alpha-1 helix of each CC domain penetrating the membrane."

We thank the reviewer for this comment. We have revised the text for grammatical and syntactical improvements where possible. However, due to word limitations (for example in the ABSTRACT) we had difficulties in changing the text and left it as it is - concise and straight to the point. We acknowledge that this might not always be the easiest to understand, but we think the current text is sufficient in reflecting the significance and importance of our findings.

Lines 154-251: The passage "We identified 20 CNLs in Arabidopsis Col-0 with putative N-

myristoylation and/or S-acylation sites at their very N-terminus (Table EV1 and Fig 1A), which we named Potentially Membrane Localized (PML) NLRs." might be clearer as: "In Arabidopsis Col-0, we identified 20 CNLs with putative N-myristoylation and S-acylation sites at their N-termini, which we have termed Potentially Membrane Localized (PML) NLRs (Table EV1 and Fig 1A)."

We have changed the sentence accordingly.

Lines 370-373: The sentence "As reported above, induced (over-)expression of active/activated PML5 strongly induced cell death and altered vacuolar morphology (Fig 1 and 4), suggests that PML5 functions as a canonical CNL but with some specific characteristics - a dynamic Golgi/tonoplast localization and the vacuolar fragmentation phenotype." could be improved for readability by adjusting it to: "As previously mentioned, the induced overexpression of active PML5 markedly induces cell death and alters vacuolar morphology (Fig 1 and 4), indicating that PML5, while functioning as a canonical CNL, also exhibits distinct characteristics, such as dynamic localization to Golgi and tonoplast, and a pronounced vacuolar fragmentation phenotype."

We thank the reviewer for bringing up this and other suggestions for phrasing some of our sentences. We have changed the sentence following the reviewer's suggestion.

Figure 5: Extended figure 9: green text and green y and x axis look awkward and light green hard to read on white screen.

We appreciate this comment. We have changed the text and axis to standard black color.

Figure 1 Legend: Units like 80 μ M sometimes written together instead of 80 μ M (SI Units style convention)

Thanks for this comment. We have double checked the text for these issues and changed it where it was necessary.

Extended Figure 1: Phylo tree text hard to read (Use pdf instead of png)

Thanks for the comment. We have improved the readability of the phylogenetic tree in the revised Extended View figure 1.

Figure 5 Legend: Line 906 has an alpha value but Line 895 does not have

Thank you for catching this error. We have added the alpha value for figure 5 panel C.

Line 934: "upper panle" -> panel

This is changed. Thank you.

Line 968: NBARC => NB-ARC

Thanks. Has been changed in the revised text.

Extended Figure 10 A: Use of 3 colors but not mentioning what their meanings are?

We now have indicated this in the figure legend itself.

Line 1047: Error in legend: (A) Pst DC3000 is wrong according to the image it's Pst DC3000 avrRPM1.

Thank you for catching this mistake. We have corrected it accordingly.

Referee #2:

Sunil and colleagues present a detailed dissection of the atypical plant immune receptor PML5 in Arabidopsis. They nicely demonstrate the autonomous activity of the CNL/ANL-type receptor and importantly demonstrate novel localization to the tonoplast/vacuole and conserved biochemical functions in oligomerization and N-terminal motif-mediated pore formation and cell death. This is a complete study with a solid and appropriate breadth of experimentation that addresses the majority of questions surrounding the evolution and function of a new/atypical immune receptor architecture. I am highly supportive of this work overall and have only a few minor comments/suggestions.

We thank this reviewer for the very positive evaluation of our work and the great appreciation of it. Thank you.

Main Comments:

- Given that wildtype PML5 causes cell death on its own, it seems relevant to explore it's transcriptional regulation during stress. Is PML5 generally upregulated during basal immune responses, ETI, or other stresses? Any correlation to stresses leading to vacuolar fragmentation/cell death?

We thank the reviewer for this comment. We have investigated published expression data sets (for example Yang et al., 2021 Front. Plant Sci.) or one from a previous publication of our lab (Saile and Jacob et al., 2019 Plos Biol) and have found that PML5 is not upregulated (potentially rather weakened) during virulent or avirulent infections at the time points analyzed in the publications (reviewer figure 1). We have also included a statement in the discussion of the revised manuscript (“However, we wanted to note that no transcriptional upregulation of PML5 was reported during virulent or avirulent bacterial infections.”).

Beside the change in vacuolar morphology induced by the PIEZO mechanoreceptor/channel protein family upon mechanic stress, we did not find any other evident correlation of a similar phenotype during abiotic or biotic stresses – at least to our knowledge.

Reviewer Figure 1. PML NLR expression during PTI and different ETIs in Col-0. No transcriptional upregulation observed for PML5 during infections with Pf0-1 EV (LFC_Col_EV), Pf0-1 AvrRpm1 (LFC_Col_M1), Pf0-1 AvrRps4 (LFC_Col_S4) and Pf0-AvrRpt2 (LFC_Col_T2). Heatmap is showing the expression value (log2 foldchange) as compared to uninfected Col-0 at 30min (30m), 4 hours (4h) and 8 hours (8h) after infection.

- Fig EV9: The false-colored images appear to show decreased HR intensity in *pml5* mutants (more so in *pml5-2c*) for CNL but not TNL-mediated responses. Did the authors explore this? Quantification of several images (pixel intensity), trypan blue staining, or ion leakage assays could be helpful here.

Thank you for this interesting comment. This difference in the false-colored images of the infiltrated leaves showing the CNL-mediated cell death responses did not occur to us previously. We therefore investigated this in more detail and used the leaf-images of the previous and new infection experiments for pixel intensity quantifications, as suggested by the reviewer. However, we did not observe a statistically significant difference in the cell death timing or strength for the CNL-mediated responses tested (see new Expanded View Figure 4).

Do you think that PML5 potentiates cell death activation to give a more robust HR? If so, are PML5 transcripts induced during interactions with virulent or avirulent Pst?

We appreciate this comment. As written in the response to the first comment of this reviewer, we were not able to detect or discover any significant upregulation of the transcript of *PML5* and thus a putative potentiation of cell death activation by other NLRs (CNLs or RNLs) seems unlikely. Further, we were not able to detect any significant decrease in cell death induced by tested CNLs and RNLs in the *pml5* mutants. Therefore, we do not think that PML5 activity potentiates cell death for the tested NLRs to give a more robust HR. However, we can of course not exclude the possibility that PML5 may contribute to 'weak' ETI responses initiated upon infection with Pst DC3000 (Laflamme et al., 2020 Science, doi: 10.1126/science.aax4079.).

Minor Comments:

-Lines 330-341: Really nice data. Do these sizes make biological sense for distinct sizes of oligomers? Are you seeing pentamers or even higher complexes? Is it possible or informative to model the oligomer using AF2?

Thank you for your appreciation. We also think that this BN-PAGE data adds significantly to our analysis. Given the size of about 110kDa for the PML5-Citrine-HA fusion protein the complexes showing up only for the cell death active PML5 proteins between 720 and 1236 kDa too big for a pentameric resistosome as modelled with AlphaFold 2 and 3 (see reviewer figure 2). The complex being observable for all PML5 fusion proteins (below 720 kDa) could fit with such a predicted pentamer but given that this is a complex showing up for the non-cell death inducing mutants as well, we doubt that it represents a functional resistosome like structure. However, we of course cannot exclude the possibility that the complex present for all PML5 fusion proteins below 720 kDa does indeed represent a pentameric resistosome. This would mean that all mutants are capable of forming such an oligomer but only the cell death active ones can be incorporated in bigger complexes required for cell death induction. This, however, is difficult to test and would for example require the identification of mutations affecting oligomerization and/ or self-association in PML5.

Given the recently preprinted hexameric and dodecameric structure of NbNRC4 it is very well conceivable that not all CNLs form pentameric resistosomes and that other higher-order oligomers are plausible. Therefore, we do also think that the structural prediction of a PML5 pentamer is not very useful for the story presented in our manuscript, beside rather good pTM (=0.61) and pIDDT values for the predictions (reviewer figure 2B right panel).

Reviewer Figure 2. PML5 pentamer prediction. A and B) Full-length PML5 protein sequence was used to predict the model of a potential PML5 pentamer with AlphaFold 2 multimer (A) and AlphaFold 3 (B). For the model in (B) the PAE blot is shown.

-Fig 4B: Tonoplast marker is hard to interpret, looks like signal everywhere. Is there a better marker for this?

We thank the reviewer for this comment. The marker used in Figure 4B for the co-localization analysis is not a tonoplast marker *per se* but rather a marker for the vacuolar lumen (*p35S::sp-RFP-AFNY-STOP-tRFP*). We generally experience that transient overexpression of this luminal RFP leads to an overloading of the trafficking system and thus we often detect RFP signal also in other endomembrane compartments – most likely the ER or pre-vacuolar compartments. Additionally, we cannot fully exclude that overexpression of PML5, which causes the severe changes in vacuolar morphology, has also an effect on the secretion of cargo and membranes to the vacuole and therefore such a luminal vacuolar mRFP marker would be visible in compartments along this secretory pathway. This was also somehow visible in the FM4-64 co-localisation analysis of which we provided an example in the revised version (Figure 4G). Here, we often observed rather big FM4-64 positive aggregates. We are now going to further investigate the possible effect of PML5 expression/function on membrane and cargo trafficking to the vacuole. We have repeated some of our co-localization analysis with a tonoplast marker (CBL6-mCherry tag) to better visualize the PML5 tonoplast localization. However, we do think that we also observe an effect on vacuolar trafficking when active PML5 is expressed, potentially because of the influence PML5 activity has on the vacuolar morphology. This observation is very interesting, and we are currently investigating this in more detail and thus do not think it is worse discussing this in the current manuscript at this preliminary stage of our analysis.

-Fig EV8 and localization data: Is there an issue localizing PML5-mCitirne-HA at the membrane in confocal experiments vs fractionation? Why do you think this is happening and is the tag interfering with the process? One suggestion is to perform localization experiments using the N-term motif mutants that lose cell death (L-to-E in N-terminal motif) but are presumably still oligomerizing as is described and exploited by NRCs. This might improve imaging.

We thank the reviewer for bringing up this issue and idea/suggestion. To avoid too much influence or disturbance of the cell death induction by imaging PML5 at the confocal microscope we used the estradiol inducible system, where we can relatively reproducibly image PML5 localization before cell death occurrence. Since, PML5-Citrine-HA is functional in inducing cell death similar to a PML5 version with a shorter epitope tag (for example HA alone or myc) we do not think that there is an issue with the localization of the Citrine-HA tagged proteins. We did the localization of our PML5 T15E,L16E,F20E (EEE) N-terminal mutant, which loses cell death activity and (strong) oligomerization, and we can also observe this protein colocalizing with a Golgi-marker demonstrating that the localization results of our WT PML5 version is representing the biologically relevant (and most likely natural) localization site – the Golgi and tonoplast.

We also wanted to note that it sometimes is difficult to distinguish between the cytosol or tonoplast in fully expanded and developed epidermal cells by 'normal' confocal microscopy due to the resolution limits of this technique. However, we are convinced that the observed (interpreted) colocalization of wild-type PML5 with a Golgi- and tonoplast marker reflects the biologically relevant and naturally also occurring localization. This is also supported by our confocal analysis using stable transgenic Arabidopsis seedlings expressing wild-type PML5-GFP.

-Line 39: typo "PML5-ike"

Thanks for catching this typo. We have corrected it.

-Fig 5D: The lines flowered early, but it would be useful to mention the timepoint that this happened at, and the relative difference to WT. Did they flower much earlier, or just a week earlier?

We appreciate this comment. We have repeated the flowering experiment again under our greenhouse conditions and observed that all plants of both *pm15* alleles flowered about 4-7 days earlier than the Col-0 wildtype plants, which was about four weeks after germination in our conditions (Col-0 started flowering at about five weeks after germination). We have now indicated the timepoint in the text and figure legend.

-Line 772: KB 'agarose' plates. Agarose is mentioned a few times and should be corrected to agar.

Thanks. We corrected this mistake accordingly.

-grammar in the title. 'An atypical...'

Thank you. We have changed the title to better reflect the findings and to better stick to the character count to: 'Activation of an atypical plant NLR with a N-terminal deletion initiates cell death at the vacuole'.

Referee #3:

This manuscript by Sunil et al provides a large amount of data that collectively support that Arabidopsis thaliana PML5, an atypical NLR with a deletion in in the CC domain, functions as a canonical NLR. Comparative genomics revealed that such a deletion in CC domain is not restricted to A. thaliana but is present in other Brassicales plants. The authors then showed that PML5 has ability to oligomerize, to increase cytosolic Ca concentration, and to induce cell death. Interestingly, the authors found the localization to Golgi and tonoplast is a unique property of PML5, because other ANL class of NLRs have been reported to localize plasma membrane via post-translational modifications such as N-myristoylation and S-acylation. Pathoassays using a T-DNA insertional mutant and a newly created CRISPR mutant support a role of PML5 in basal immunity against Pseudomonas syringae.

Overall, the conclusion of this manuscript is supported well by the solid data, and the writing is clear and easy to follow. The novelty resides in the discovery of the unique mode of cell death induction by PML5. I have only a few comments that may further improve this nice manuscript.

We thank the reviewer for the positive evaluation of our manuscript and the precise summary of our results.

Major points:

1. The authors argue that PML5 induces vacuolar fragmentation that correlate with cell death induction. I think this claim needs a different line of evidence or through examination.

This is a valid concern, and we thank the reviewer for bringing this up. However, we do also want to point out that we only describe a correlation between cell death induction by PML5 and the observed vacuolar fragmentation, which we only see if cell death active PML5 variants are expressed. We agree that at this state it is not clear if vacuolar fragmentation is (i) caused directly by PML5 function/activity, or (ii) whether it is the cause or consequence of PML5 induced cell death. We have performed some preliminary experiments with other cell death inducing NLRs (or domains of these – see below) and do not observe a similar effect on vacuolar morphology shortly before the onset of macro- and microscopically visible cell death.

In Fig. 4B, the localization pattern of the vacuole marker protein is unaltered. How can this be explained?

We honestly have to respond that we do not fully understand the concern raised here. We apologize if we do not fully respond to this concern. We would not have expected a dramatic change in the localization pattern of this marker during PML5 expression.

It is true that in Fig 4B the luminal vacuolar mRFP fluorescence is still visible in the fragmented vacuoles of the PML5 expressing cell. However, we do observe that in the smaller fragmented vacuoles of these cells the intensity of the fluorescence is weaker, and it seems that stronger fluorescence is observed in some structures or aggregates, also positive for PML5 fluorescence, visible at the borders of the cell in the specific image. These may represent some endomembrane compartment that are of the secretory pathway to the vacuole (prevacuolar compartments, multivesicular bodies), but this requires further analysis.

Is vacuolar fragmentation specific to PML5-induced cell death? How can you rule out the possibility that the vacuolar fragmentation is an artefact due to PML5 overexpression? Have you tested whether PML5 driven by native promoter induces vacuolar fragmentation? It could also be that vacuolar fragmentation is a consequence of cell death not an active regulation by PML5.

These are very good and important questions, and we are currently investigating these in more detail in the lab. However, we do think that the answers or the attempts (experiments) to answer them are beyond the studies or scope of this study, which is a first characterization of this (in our opinion) very interesting and atypical NLR. We do have some preliminary data on these questions that we would like to share here with the reviewers and, if published, the community in the form of the reviewer process file (RFP). Our preliminary experiments strongly indicate that this vacuolar fragmentation phenotype is indeed specific for PML5 induced cell death or the expression of cell death active PML5 (see reviewer figure 3). We did not observe a similar vacuolar fragmentation or change in morphology when we coexpressed our luminal vacuolar marker with an autoactivated mutant PML11^{DV}/At1g61180 (Beeh et al. unpublished), RPM1^{DV} (El Kasmi et al., 2017 PNAS) or the cell death inducing CC-NB-ARC domain of ADR1-L2 (Saile et al. unpublished) or CC domain of ADR1 (Saile et al., 2021 New Phytologist). That the PML5 induced or correlating fragmentation of the vacuole is an artefact of overexpression is difficult to exclude. However, we do observe cell death induction by

native promoter driven expression of wildtype and D362V mutant PML5 when heterologous expressed in *N. benthamiana*, and we thus assume to see a similar change in vacuolar morphology by confocal analysis of the luminal marker, but we had thus far difficulties in getting enough expression of PML5 to see its fluorescence in such an experiment. Therefore, we cannot convincingly say that native promoter driven PML5 would also induce such a vacuolar phenotype.

Taken together, we do not think that vacuolar fragmentation is a consequence of cell death in general, but rather a direct consequence of PML5 acting on the tonoplast, very likely forming resistosome-like structures leading to calcium influx to the cytosol out of the vacuole.

Reviewer Figure 3. No observable change in vacuolar morphology upon expression and cell death induction of auto-activated NLRs or NLR domains. MHD mutant auto-active versions of PML11 and RPM1, as well as auto-active ADR1-L2 and ADR1 domains were transiently co-expressed together with a luminal vacuolar mRFP marker in *N. benthamiana*. Expression was induced 18-24 hours post infiltration by application of 20 μ M estradiol. Imaging was done between 5-18 hours post induction.

2. Are the GCC residues required for PML5 localization to Golgi membrane and the tonoplast? No data are provided testing colocalization of GCC(2, 3,4)A-Ct-HA and the Golgi or Vacuole marker

proteins.

We appreciate this comment. We have now included an experiment where we did co-localization of the GCC(2,3,4)A-Cit-HA mutant with our newly cloned tonoplast marker (see new figure EV3I). We do see some co-localization in our experiments but also a strong localization of the mutant protein in the cytosol. We thus cannot fully rule out if the mutation of the N-myristoylation and S-acylation sites is sufficient to fully abolish membrane localization of PML5. Also, because it is very difficult to distinguish between the cytosol and the tonoplast in these images, given the very close proximity of the cytosol and the tonoplast in fully expanded and developed leaf epidermal cells.

We do want to mention that there are other cysteine residues in the PML5 protein sequence within a predicted S-acylation motif, which very well could contribute to proper membrane localization. These can be also found in other PMLs and have been shown to be important for the rice Pit CNL and the *N. benthamiana* RSS1 (Kawano et al., 2014 - <https://doi.org/10.1074/jbc.M114.569756>; Kim et al. 2023 - <https://doi.org/10.1016/j.xplc.2023.100640>). We are currently investigating the importance of these residues for PML5 and PML11 function in another project in the lab.

Other comments:

1. Is PML5 transcriptionally induced upon pathogen challenge? If this is the case, it would support the role of PML5 in immunity, as transcriptional induction of PML5 is sufficient to induce cell death (Fig. 1E and Fig EV3E).

We appreciate this comment, which was also brought up by reviewer 2. We could not find any indication that PML5 transcription is induced upon bacterial infiltration (virulent and avirulent) – see also response to main comment 1 of reviewer #2.

Along the same line, have you tested whether pre- or co-treatment of estradiol with *P. syringae* enhances bacteria resistance in the transgenic lines used in Fig EV4E?

This is an interesting comment and a great question. At this time, we did not do this experiment, but want to investigate this further in the future. This is because we could not observe any PR1 (pathogenesis response 1) protein accumulation (by protein blot) in our PML5 induced over-expressor *Arabidopsis thaliana* lines at 6, 13 and 20 hours post estradiol induction and only a weak induction of *PR1* transcript in qRT-PCR analysis at 6 hours post estradiol induction (all preliminary results). Therefore, it would be very interesting to see what the effect of induction of *PML5* expression and the resulting cell death is on *P. syringae* infections.

2. Fig EV6: RPMI is labeled 0/20, although I see clear cell death in the infiltrated area.

Thank you for pointing this mistake out. We have now changed this to (20/20) in the revised manuscript (old EV figure 6 new EV figure 2).

3. Fig EV10: Statistical analysis should be applied.

We thank the reviewer for mentioning this issue. We now have included the statistical analysis in the figure and legend, although there was no statistical difference detected between the mutant and the wildtype.

Dear Dr. El Kasmi,

Thank you for the submission of your revised manuscript to our editorial offices. I have now received the reports from the three referees that I asked to re-evaluate the study, you will find below. As you will see, the referees now support the publication of the study in EMBO reports. Referees #2 and #3 have some remaining concerns and suggestions to improve the manuscript, I ask you to address in a final revised manuscript. Please also provide a final p-b-p-response regarding the remaining points of the referee.

- Please provide the abstract written in present tense.
- Please make sure that the number "n" for how many independent experiments were performed, their nature (biological versus technical replicates), the bars and error bars (e.g. SEM, SD) and the test used to calculate p-values is indicated in the respective figure legends. Please also check that all the p-values are explained in the legend, and that these fit to those shown in the figure. Please provide statistical testing where applicable. Please avoid the phrase 'independent experiment', but clearly state if these were biological or technical replicates. Please also indicate (e.g. with n.s.) if testing was performed, but the differences are not significant. In case n=2, please show the data as separate datapoints without error bars and statistics. See also: <http://www.embopress.org/page/journal/14693178/authorguide#statisticalanalysis>

If n<5, please show single datapoints for diagrams. Presently, some diagrams seem to miss the 'n.s.'. Please check. Moreover:

- Please note that the figure 4a-e is mislabeled as figure 4a-d in the manuscript. This needs to be rectified.
- Please note that the figure EV 4a-m is mislabeled as figure EV 4a-l in the manuscript. This needs to be rectified.
- Please define the annotated p values a/b/c as well as provide the exact p-values for the same in the legend of figure 5f; EV 4a, c-d, f-g, i-j, l-m; as appropriate.
- Please note that the exact p values are not provided in the legend of figure EV 5a.
- Please note that the p value "****" is not represented in the figure EV5b, however statistical test related information is provided in the legend of the corresponding figure. This needs to be rectified. Also, provide the exact p-value for the same.
- Please note that the box plots need to be defined in terms of minima, maxima, centre, bounds of box and whiskers, and percentile in the legends of figures 5c; EV 4a, c-d, f-g, i-j, l-m; EV 5a-c.
- Please add to each legend (main, EV and Appendix figures, where applicable) a 'Data Information' section explaining the statistics used or providing information regarding replicates and scales. See:

- Per journal policy, we do not allow 'data not shown', which is stated in the manuscript (page 12). All data referred to in the paper should be displayed in the main or Expanded View figures, or an Appendix. Thus, please add these data (or change the text accordingly if these data are not central to the study). See: <https://www.embopress.org/page/journal/14693178/authorguide#unpublisheddata>

- Please make sure that all the funding information is also entered into the online submission system and that it is complete and similar to the one in the acknowledgement section of the manuscript text file. The University of Tübingen needs also be entered as a separate funder in the submission system.

- Please make sure that all figure panels are called out separately and sequentially. Presently, there seems to be no callout for panel 1G. Please check.

In addition, I would need from you:

Please use this link to submit your revision: <https://embor.msubmit.net/cgi-bin/main.plex>

Best,

Referee #1:

Sunil et al. have thoroughly addressed all comments and have incorporated the suggested changes. The manuscript has undergone significant revisions, and the language has improved notably. I have no further suggestions for the authors.

Referee #2:

The authors present a thoroughly revised and improved manuscript from what was already an excellent work.

I only have 1 minor point that needs attention. On Line 470: suggests that conserved residues in the primary alpha helix are important for oligomerization, whereas the Leucine residues (or even the full primary helix) from cell death inducing CNLs/RNLs appear to only impact cell death induction without impacting oligomerization in BN-PAGE experiments. MADA mutants of NRC2 can still oligomerize, as well as the N16 truncated version of NRG1 (examples below).

<https://www.ncbi.nlm.nih.gov/pmc/articles/PMC9975942/>

<https://www.embopress.org/doi/full/10.15252/emboj.2022111519>

<https://www.science.org/doi/full/10.1126/science.abg7917>

Referee #3:

I apologize my unclear question about the vacuolar marker protein. I raised the same concern as reviewer 2, and in the revised manuscript the authors have nicely addressed this concern by providing data for colocalization of a tonoplast marker and PML5. I am essentially satisfied with the revised manuscript, but please consider the following comments before publication.

1. Fig 4B: It was very difficult for me to spot vacuolar fragmentation based on the vacuolar-mRFP signal alone. So, please consider providing enlarged images of the area where vacuolar fragmentation takes place?
2. Since the cell death-defective GCC(2,3,4) and EEE mutants still showed localization to the Golgi and the tonoplast (EV Fig 3I and K), the importance of the Golgi and tonoplast localization of PML5 for the execution of cell death is not fully supported. So, the authors' claim about this should be toned down such as those in lines 401 and 481. For example, in line 401 "seemed to be" should be changed to "may be".

Dear Achim,

Please find below a point-by-point (p-b-p) response to your and the reviewers requests. We have also included a short summary, highlighting bullet points and a graphical summary. Thank you for your professional and kind handling of our manuscript.

Best,
Farid

p-b-p Responses to the editorial and reviewer requests:

- Please provide the abstract written in present tense.
We now have changed the abstract to be in present tense.

- Please make sure that the number "n" for how many independent experiments were performed, their nature (biological versus technical replicates), the bars and error bars (e.g. SEM, SD) and the test used to calculate p-values is indicated in the respective figure legends.
We now have included the requested information in the figure legends where required to the best of our knowledge.

Please also check that all the p-values are explained in the legend, and that these fit to those shown in the figure.

All p-values are now explained in the figure legends where required.

Please provide statistical testing where applicable.

Statistical testing is now provided where we think its necessary and important.

Please avoid the phrase 'independent experiment', but clearly state if these were biological or technical replicates.

We now have clearly stated the 'nature' of the replicates in the figure legends.

Please also indicate (e.g. with n.s.) if testing was performed, but the differences are not significant.

We did indicate not significant differences with n.s. in the figures where necessary.

In case n=2, please show the data as separate datapoints without error bars and statistics.

We do not show any blot with n=2.

If n<5, please show single datapoints for diagrams. Presently, some diagrams seem to miss the 'n.s.'. Please check.

Where it is relevant we do show all data as single datapoints. We also have included the 'n.s.' in relevant blots.

Moreover:

- Please note that the figure 4a-e is mislabeled as figure 4a-d in the manuscript. This needs to be rectified.

Thanks. We changed it accordingly.

- Please note that the figure EV 4a-m is mislabeled as figure EV 4a-l in the manuscript. This needs to be rectified.

Thank you. The labeling was corrected.

- Please define the annotated p values a/b/c as well as provide the exact p-values for the same in the legend of figure 5f; EV 4a, c-d, f-g, i-j, l-m; as appropriate.

We now have included all the necessary information in the respective figure legends.

- Please note that the exact p values are not provided in the legend of figure EV 5a.

We now have provided the p-values for this figure panel.

- Please note that the p value """" is not represented in the figure EV5b, however statistical test related information is provided in the legend of the corresponding figure. This needs to be rectified. Also, provide the exact p-value for the same.

We now have changed the specific figure and figure legend to represent the required information.

- Please note that the box plots need to be defined in terms of minima, maxima, centre, bounds of box and whiskers, and percentile in the legends of figures 5c; EV 4a, c-d, f-g, i-j, l-m; EV 5a-c.

We now have included all the necessary description/definition of our box plots in the specific figure legends.

- Please add to each legend (main, EV and Appendix figures, where applicable) a 'Data Information' section explaining the statistics used or providing information regarding replicates and scales.

We have included the necessary Data information section in figure legends where required.

- Per journal policy, we do not allow 'data not shown', which is stated in the manuscript (page 12). All data referred to in the paper should be displayed in the main or Expanded View figures, or an Appendix. Thus, please add these data (or change the text accordingly if these data are not central to the study).

We have changed the text accordingly.

- Please make sure that all the funding information is also entered into the online submission system and that it is complete and similar to the one in the acknowledgement section of the manuscript text file. The University of Tübingen needs also be entered as a separate funder in the submission system.

The University of Tübingen is entered as a separate funder now.

- Please make sure that all figure panels are called out separately and sequentially.

Presently, there seems to be no callout for panel 1G. Please check.

Thanks. Was mistakenly labeled as Fig 1E and not G.

In addition, I would need from you:

- a short, two-sentence summary of the manuscript (not more than 35 words).

- two to four short (!) bullet points highlighting the key findings of your study (two lines each).

- a schematic summary figure as separate file that provides a sketch of the major findings (not a data image) in jpeg or tiff format (with the exact width of 550 pixels and a height of not more than 400 pixels) that can be used as a visual synopsis on our website.

We have now included a short, two-sentence summary, highlighting bullet points and a schematic summary.

Referee #1:

Sunil et al. have thoroughly addressed all comments and have incorporated the suggested changes. The manuscript has undergone significant revisions, and the language has improved notably. I have no further suggestions for the authors.

Thank you for the kind words and the recognition of the work we put into the revision.

Referee #2:

The authors present a thoroughly revised and improved manuscript from what was already an excellent work.

Thank you very much for this enthusiastic comment.

I only have 1 minor point that needs attention. On Line 470: suggests that conserved residues in the primary alpha helix are important for oligomerization, whereas the Leucine residues (or even the full primary helix) from cell death inducing CNLs/RNLs appear to only impact cell death induction without impacting oligomerization in BN-PAGE experiments. MADA mutants of NRC2 can still oligomerize, as well as the N16 truncated version of NRG1 (examples below).

<https://www.ncbi.nlm.nih.gov/pmc/articles/PMC9975942/>
<https://www.embopress.org/doi/full/10.15252/emboj.2022111519>
<https://www.science.org/doi/full/10.1126/science.abg7917>

We thank the reviewers for the comment. We now rephrased this conclusion and the section in the results to reflect this. It is true that mutating the conserved leucine (and other conserved hydrophobic) residues is not affecting activated (by effector) NRC2 or NRC4. We did not test if our PML5 T15A,L16A,F20A in the DV context still oligomerizes, which could be the case. We did sometimes observe a weak oligomerization of the G,C,C, DV PML5 variant in our BN PAGE assays, but this was only visible when the blot was exposed for very long (about 20min). We assume that PML5 TLF to A and GCC to A mutants lose their ability to oligomerize efficiently in absence of an auto-activation, for example by the DV mutation. It is weakly visible on the BN-PAGE blot in Figure 3E that the GCC mutants (in wt and DV background) still do form a complex above the 720 kDa size.

Referee #3:

I apologize my unclear question about the vacuolar marker protein. I raised the same concern as reviewer 2, and in the revised manuscript the authors have nicely addressed this concern by providing data for colocalization of a tonoplast marker and PML5. I am essentially satisfied with the revised manuscript, but please consider the following comments before publication.

We thank the reviewer for the great and professional evaluation of our revised manuscript and all the valuable input.

1. Fig 4B: It was very difficult for me to spot vacuolar fragmentation based on the vacuolar-mRFP signal alone. So, please consider providing enlarged images of the area where vacuolar fragmentation takes place?

Thank you for this suggestion. We now have marked a representative area in the RFP panel of figure 4B to highlight/indicate the observed fragmentation. We decided to not show a zoomed in additional image because of the esthetics/symmetry of the whole figure and because we think that with our indication it will be obvious what we mean.

2. Since the cell death-defective GCC(2,3,4) and EEE mutants still showed localization to the Golgi and the tonoplast (EV Fig 3I and K), the importance of the Golgi and tonoplast localization of PML5 for the execution of cell death is not fully supported. So, the authors' claim about this should be toned down such as those in lines 401 and 481. For example, in line 401 "seemed to be" should be changed to "may be".

Thank you for this comment. We have toned down the relevant statements for the EEE mutant accordingly. However, we believe the GCC mutant actually is affected in proper Golgi and tonoplast localization, since we do not observe any punctate localization and the co-localization profile indicates that the fluorescence peaks do not fully overlay for the GCC mutant and the tonoplast marker (see figure below and in profile blot of figure EV3 I.).

Dr. Farid El Kasmi
Eberhard Karls University of Tübingen
Center for Plant Molecular Biology
Auf der Morgenstelle 32
TÄ1/4bingen, BW 72076
Germany

Dear Dr. El Kasmi,

I am very pleased to accept your manuscript for publication in the next available issue of EMBO reports. Thank you for your contribution to our journal.

Yours sincerely,
